# The Renaissance of Classic Feature Aggregations for Visual Place Recognition in the Era of Foundation Models

## Abstract

Visual Place Recognition (VPR) addresses the retrieval problem in large-scale geographic image databases through feature representations. Recent approaches have leveraged visual foundation models and have proposed novel feature aggregations. However, these methods have failed to grasp the core concepts of foundational models, such as leveraging extensive training sets, and have also neglected the potential of classical feature aggregations, such as GeM and NetVLAD, for low-dimensional representations. Building on these insights, we revive classic aggregation methods and create more fundamental VPR models, abbreviated SuperPlace. First, we introduce a supervised label alignment method that combines grid partitioning and local feature matching. This allows models to be trained on diverse VPR datasets within a unified framework, similar to the design principles of foundation models. Second, we introduce $G^2M$, a compact feature aggregation with two GeMs, in which one GeM learns the principal components of feature maps along the channel direction and calibrates the other GeM's output. Third, we propose the secondary fine-tuning ($FT^2$) strategy for NetVLAD-Linear (NVL). NetVLAD first learns feature vectors in a high-dimensional space and then compresses them into a low-dimensional space using a single linear layer. $G^2M$ excels in large-scale applications requiring rapid response and low latency, while NVL-$FT^2$ is optimized for scenarios demanding high precision across a broad range of conditions. Extensive experiments (12 test sets, 14 previous methods, and 11 tables) highlight our contributions and demonstrate the superiority of SuperPlace. Specifically, SuperPlace-$G^2M$ achieves state-of-the-art results with only one-tenth of the feature dimensions compared to recent methods. Moreover, SuperPlace-NVL-$FT^2$ holds the top rank on the MSLS challenge leaderboard. We have submitted a ranking screenshot, the source code, and the original experimental records in the supplementary materials.

## 1 Introduction

Visual Place Recognition (VPR), also known as Visual Geo-localization, involves finding the most similar image to a query image within a large-scale geographic image database (Berton et al., 2022b). VPR has long been studied in computer vision (Sattler et al., 2018), robotics (Lowry et al., 2015), and remote sensing (Psomas et al., 2024) due to its wide applications in augmented reality, robot navigation, and autonomous driving. Previous research has identified several challenges in VPR, including large database scales (Berton et al., 2022a), viewpoint shifts (Berton et al., 2021a), repeated structures (Torii et al., 2015), structural modifications (Arandjelovic et al., 2016), occlusions (Liu et al., 2021), visual scale differences (Fu et al., 2022), illumination changes (Liu et al., 2024), and seasonal transitions (Toft et al., 2020).

**Early** VPR research aggregated hand-crafted local features such as SURF (Bay et al., 2008) into global features through algorithms like Bag-of-Words (BoW) (Angeli et al., 2008) or Vector of Locally Aggregated Descriptors (VLAD) (Jégou et al., 2010). However, most challenging problems are difficult to solve within the framework based on hand-crafted features.

**In the past decade**, VPR primarily leveraged location datasets and neural networks with differentiable aggregation/pooling to map images into an embedding space, effectively distinguishing images from different locations (Arandjelovic et al., 2016). New VPR datasets have been continually proposed to overcome challenging problems (such as generalization (Ali-bey et al., 2022), training scale (Berton et al., 2022a), and cross-domain (Warburg et al., 2020)). However, these datasets do not fully encompass the characteristics of previous datasets and introduce a new issue: inconsistent supervised label formats (Berton et al., 2022a). At the same time, different methods have also been proposed to aggregate multi-channel feature maps into global feature vectors. Generalized Mean Pooling (GeM) and NetVLAD (Arandjelovic et al., 2016) were proposed, which achieved good results in different dimension ranges (less than 4k dimensions or larger than 30k dimensions). The feature dimension strongly correlates with the retrieval speed and recall of VPR.

**In the past year**, Visual Foundation Models (VFMs) (Oquab et al., 2023; Kirillov et al., 2023; Yang et al., 2024; Wang et al., 2024) have developed rapidly, and DINOv2 has been widely used in VPR (Keetha et al., 2023; Lu et al., 2024b). VFMs have utilized multiple large-scale visual datasets, knowledge distillation (Hinton et al., 2015), and other techniques to provide powerful feature representation capabilities. However, these VPR studies have only used a VFM as a pretrained model on a single dataset and have not embraced the core principles of VFMs, such as aligning multiple datasets for model training. Additionally, these studies also proposed some novel feature aggregation methods: BoQ (Ali-bey et al., 2024), SALAD (Izquierdo & Civera, 2024a), and SPGM (Lu et al., 2024a). The feature dimensions of these methods were approximately 4k-12k. They were expected to perform better than GeM, and these works claimed to produce better results with lower feature dimensions than NetVLAD. However, through some tentative experiments, we found that classic methods from ten years ago are still competitive. This prompted us to improve these classic methods instead of following recent aggregations introduced in the past year.

**In this paper**, we not only use VFMs as pre-trained models but also train on multiple datasets, mirroring their training approach through a novel supervised alignment method. In particular, we transform the distance metric into class labels through a grid partition in the Universal Transverse Mercator (UTM) coordinate and check the similarity within the labels using local feature matching. Beyond staying updated with the latest techniques, we also improve two feature aggregations from over a decade ago to revive their superiority in the era of VFMs. Specifically, we propose a generalized channel attention module for GeM and the secondary fine-tuning ($FT^2$) for NetVLAD-Linear (NVL). *With the same training set*, these improved aggregations achieve comparable or even superior results to recent approaches while requiring *lower dimensions and fewer parameters*. Our contributions are highlighted as follows:

1) We propose a supervised label alignment method to train VPR models using multiple datasets like other foundation models. Specifically, the coarse classification labels are first determined by a grid partition in the UTM coordinate, and then fine labels are selected by using local feature matching.

2) We propose a compact feature aggregation with two GeM pooling layers, $G^2M$, in which one GeM learns the principal components of feature maps along the channel direction and calibrates the other GeM's output.

3) We propose the secondary fine-tuning method for NetVLAD-Linear, called NVL-$FT^2$, which first learns feature representations in high-dimensional space and then compresses the representations into low-dimensional space using a single linear layer.

4) Extensive comparative and univariate experiments demonstrate our contributions and the excellence of SuperPlace. SuperPlace-$G^2M$ achieves state-of-the-art (SOTA) results using only one-tenth of the feature dimensions of recent methods. SuperPlace-NVL-$FT^2$ holds the top rank on the MSLS Challenge leaderboard, significantly outperforming recent methods.

## 2 RELATED WORK

Early VPR research primarily relied on hand-crafted features, including global features extracted directly (like GIST (Milford & Wyeth, 2012)) and features derived from clustering local descriptors. Clustering algorithms such as BoW (Angeli et al., 2008), Fisher Vector (FV) (Csurka & Perronnin, 2010), and VLAD (Jégou et al., 2010) were used in conjunction with local feature extraction algorithms like SIFT (Lowe, 2004), SURF (Bay et al., 2008), and ORB (Rublee et al., 2011).

With the advent of deep learning, learning-based features have largely supplanted hand-crafted features. Arandjelovic et al. (2016) introduced the Pittsburgh-250k dataset along with a differentiable VLAD aggregation module and optimized a pre-trained (PT) model using a triplet loss function to achieve VPR. NetVLAD (Arandjelovic et al., 2016) laid the foundation for learning-based VPR.

**Training sets.** VPR training sets were primarily obtained from images captured by Google Street View (GSV) (Anguelov et al., 2010) and car-mounted cameras. Unlike Pittsburgh-250k, which was collected using similar cameras, MSLS (Warburg et al., 2020) was gathered from different cameras and included challenging scenes such as variations in weather, seasons, and lighting. Berton et al. (2022a) proposed the SF-XL dataset containing millions of images for training and verifying VPR models in large-scale scenarios. GSV-Cities (Ali-bey et al., 2022) and SF-XL were concurrently developed datasets based on GSV. GSV-Cities contained more diverse urban samples but had relatively sparse samples, while SF-XL comprehensively covered the street scenes of San Francisco. There is still no consensus on the best training set, and almost no studies have utilized multiple datasets. To our knowledge, SALAD-CM (Izquierdo & Civera, 2024b) is the only method, apart from ours, that uses multiple datasets (GSV and MSLS).

**Aggregation Layers.** Like NetVLAD, the other classic aggregation algorithms (Hou et al., 2018; Peng et al., 2021) have been transformed into differentiable modules for end-to-end training. Although these NetVLAD-inspired modules have demonstrated good performance, their high-dimensional characteristics limit database size and retrieval efficiency. Generalized Mean (GeM) pooling (Radenović et al., 2018) was introduced as a simpler alternative to NetVLAD, providing low-dimensional global features. This method extends global average pooling by using the $p$-norm of local features. Recently, three aggregation modules have been proposed: Bag-of-Queries (BoQ), SALAD, and SPGM. BoQ (Ali-bey et al., 2024) employed distinct learnable global queries to probe the input features through cross-attention, ensuring consistent information aggregation. SALAD (Izquierdo & Civera, 2024a) redefined the soft assignment of local features in NetVLAD as an optimal transport problem and employed the Sinkhorn algorithm to solve it. SPGM (Lu et al., 2024a) applied a spatial pyramid to divide feature maps at multiple levels and then used GeM pooling. Respectively, their optimal feature dimensions when used with DINOv2 are 12,288 for BoQ, 8,448 for SALAD, and 4,096 for SPGM. Unlike recent works, we make minor improvements to demonstrate the effectiveness of earlier approaches.

**Pre-trained Models.** As in most vision tasks, pre-trained (PT) models in VPR have evolved from convolutional neural networks (including residual networks) to transformers. Before 2020, works such as NetVLAD and SFRS (Arandjelovic et al., 2016; Ge et al., 2020) used VGG networks as PT models. In the past three years, CosPlace, MixVPR, and EigenPlaces (Berton et al., 2022a; Ali-bey et al., 2023; Berton et al., 2023) adopted ResNet as their backbone architecture. Recently, Any-Loc introduced DINOv2 without fine-tuning for VPR. Subsequently, SelaVPR (Lu et al., 2024b), SALAD (Izquierdo & Civera, 2024a), CricaVPR (Lu et al., 2024a), and BoQ (Ali-bey et al., 2024) adopted DINOv2 and fine-tuned it on the GSV-Cities dataset. The use of VFMs in VPR has been limited compared to other vision tasks, compared to other vision tasks, such as segmentation (Segment Anything (Kirillov et al., 2023)), depth estimation (Depth Anything (Yang et al., 2024)), and 3D reconstruction (DUST3R (Wang et al., 2024)). Our supervised alignment method enables VPR to effectively leverage multiple datasets, thereby advancing the field in line with the latest developments in VFMs.

## 3 METHODOLOGY

In this section, we first present G$^2$M, a super-compact feature extraction method designed for large-scale environments and scenarios with highly real-time requirements. Next, we present NVL-FT$^2$, an aggregation suite for general-scale applications where high performance is the priority. Finally, we describe a supervised label alignment method specifically tailored for VPR.

As illustrated in Fig. 1, we used DINOv2 to extract serialized patch tokens and the CLS tokens. The patch tokens were reshaped into a $C \times H \times W$ feature map, where $C, H, W$ represent the number of channels, the height, and the width of the feature map, respectively. For the loss function, we adopted the multi-similarity loss (Wang et al., 2019), as used in prior work (Ali-bey et al., 2023; Izquierdo & Civera, 2024a; Lu et al., 2024a).

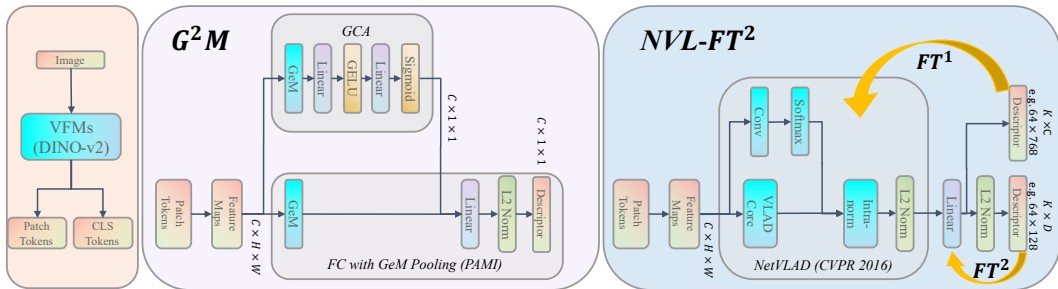

Figure 1: **Illustration of two improved classical aggregations.** Without all the bells and whistles, we improved on classic aggregations by adding simple structures, making them better than recent complex aggregation layers with many parameters.

### 3.1 GENERALIZED CHANNEL ATTENTION FOR GEM

We initially adopted the extractor proposed in Radenović et al. (2018) to generate compact feature representations. This extractor comprises Generalized Mean (GeM) pooling, a fully connected layer, and L2 normalization. The GeM pooling function is formulated as follows:

$$f = [f_1 \cdots f_c \cdots f_C], f_c = \left( \frac{1}{|X_c|} \sum_{x \in X_c} x^{p_c} \right)^{\frac{1}{p_c}}, \tag{1}$$

where max pooling and average pooling are special cases of GeM pooling. Specifically, max pooling occurs when $p_c \to \infty$, while average pooling occurs when $p_c = 1$. The pooling parameter $p_c$ is learned from each feature map.

Despite its effectiveness, this extractor is limited in its ability to fully capture the valuable information of the multi-channel feature map. To further explore this limitation, we applied PCA to reduce the channel dimension and visualized the resulting feature maps to assess their interpretability. As shown in Fig. 2, location-dependent information tends to generate strong responses, while location-independent information may be overemphasized or overlooked.

To address the above limitation and inspired by our visualizations, we introduce an additional branch that learns the principal components of the feature map along the channel dimension to calibrate the GeM pooling vector accordingly. As shown in Fig. 1, this branch consists of a new GeM pooling layer, a low-rank MLP, a GELU activation, and a Sigmoid function. This kind of simple module structure has contributed to the success of methods like the Squeeze-and-Excitation (SE) module (Hu et al., 2018) and Low-Rank Adaptation (LoRA) (Hu et al., 2021). Notably, our motivation, usage, and design details differ from those of the SE module, and we refer to this new module as the Generalized Channel Attention (GCA) module. Together, the original extractor and the GCA module form the improved extractor, which we call G²M.

### 3.2 SECONDARY FINE-TUNING FOR NETVLAD-LINEAR

NVL-FT² represents an incremental improvement over NetVLAD, whose output feature dimension is defined as $C \times K$, where $K$ denotes the number of cluster centers. In previous works, $K$ has been set to 64 in Arandjelovic et al. (2016) and 32 in Izquierdo & Civera (2024a). However, the global features extracted by NetVLAD are characterized by excessively high dimensions, prompting earlier studies to investigate two primary methods for dimension reduction: 1) employing PCA for dimension reduction, or 2) reducing the value of $K$. While the first approach introduces additional storage overhead and increased computational requirements, the second results in a substantial performance degradation.

An alternative and simpler strategy is to follow NetVLAD with a linear projection layer for dimension reduction. This method promises reduced storage requirements and faster processing times compared to PCA. Despite these theoretical advantages, our implementation of NV-Linear consistently underperformed relative to NetVLAD-PCA. This might explain why it has not been adopted or proposed in prior work.

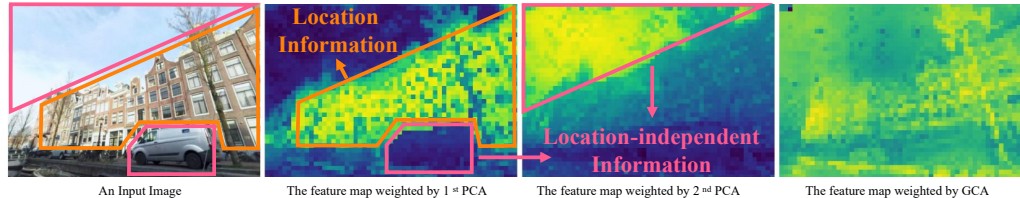

An Input Image  The feature map weighted by 1$^{st}$ PCA  The feature map weighted by 2$^{nd}$ PCA  The feature map weighted by GCA

Figure 2: **Visualization of feature maps weighted by different components.** We computed a PCA between the patches of the images from the AmsterTime dataset and showed their first three components. We found that high and low response areas of feature maps after principal component weighting strongly correlate with the VPR task.

Given that both methods operate with the same feature dimension, training set, and neural network architecture, such a performance gap is unexpected. Intuitively, the linear projection should outperform PCA. The key difference, however, lies in their training methodologies. NetVLAD-PCA employs a two-stage training procedure: (1) fine-tuning the backbone network and NetVLAD in a high-dimensional space, and (2) estimating an unsupervised model for high-to-low dimensional projection, during which the parameters from the first step are frozen. In contrast, NetVLAD-Linear utilizes a single-stage end-to-end training process, where the backbone network, NetVLAD, and the linear layer are fine-tuned simultaneously in a lower-dimensional space. This training discrepancy limits the ability of NetVLAD-Linear to capture the rich high-dimensional representations.

To overcome this issue, we propose a secondary fine-tuning process for NetVLAD-Linear. In this approach, we first fine-tune the backbone network and NetVLAD, followed by a second stage where we fine-tune the linear layer for dimension reduction. Importantly, the number of parameters involved in FT$^2$ is minimal—accounting for just 0.11% of the entire model's parameters, as noted in our experiments. Consequently, FT$^2$ is computationally efficient and enables faster training.

### 3.3 SUPERVISED LABEL ALIGNMENT FOR VPR

As mentioned above, many datasets have been proposed in the VPR field, but it is unclear whether they collectively provide comprehensive performance coverage. The difficulty in using them together lies in their different supervision labels. In this paper, we consider supervised label alignment of four widely used large-scale datasets: GSV-Cities (Ali-bey et al., 2022), Pittsburgh-250k (Arandjelovic et al., 2016), MSLS(Wang et al., 2019), and SF-XL(Berton et al., 2022a), as shown in Tab. 2. We also recorded the number of images in each dataset after aligning the labels. While SF-XL and MSLS can be further expanded (but with high redundancy), the number of images in Pitts-250k is limited by the strategy described below.

**GSV-Cities (G).** Among the datasets, GSV-Cities serves as a foundational dataset due to its recent performance (Izquierdo & Civera, 2024a). Therefore, we retain the original labels of GSV-Cities and further determine the goal, which is to convert the distance metric labels into class labels (Place IDs).

**SF-XL (S).** Following Berton et al. (2022a), we split the UTM coordinates $\{east, north\}$ of SF-XL into square geographic cells and then further divide each cell into a set of classes based on the direction/heading $\{heading\}$ of each image. Formally, the set of images assigned to the class $L_{e_i,n_j,h_k}$ would be

$$\{x : \left\lfloor \frac{east}{M} \right\rfloor = e_i, \left\lfloor \frac{north}{M} \right\rfloor = n_j, \left\lfloor \frac{heading}{\alpha} \right\rfloor = h_k\}, \tag{2}$$

where $M$ (in meters) and $\alpha$ (in degrees) are two parameters that determine the extent of each class in position and heading. $M$ is set to 10, $\alpha$ is set to 30° (Berton et al., 2022a). We also introduced CosPlace's N × L group strategy to overcome quantization errors, with N and L set to 5 and 2, respectively.

**Pittsburgh-250k (P)** has no orientation information like SF-XL. On the other hand, different slices of panoramic images should not be classified into the same category. Therefore, we design the

Table 1: Comparison of various VPR training sets.

| Dataset | # Img | # Img (SLA) | Source | Supervision | Loss Function | Related Work |
|---------|-------|-------------|--------|-------------|---------------|--------------|
| Pittsburgh | 250k | 2k | GSV | UTM | Triplet Loss | NetVLAD (CVPR16), DHEVPR (AAAI24), SelaVPR (ICLR24) |
| MSLS | 1.68M | 820k | Mapillary | UTM | Triplet Loss | TransVPR (CVPR22), R2Former (CVPR23), SALAD-CM (ECCV24) |
| SF-XL | 5.6M | 180k | GSV | UTM & Orient. | LMC-Loss | CosPlace (CVPR22), EigenPlaces (ICCV23), NocPlace (arXiv24) |
| GSV-Cities | 530k | 530k | GSV | Place ID | MS-Loss | MixVPR (WACV23), CricaVPR (CVPR24), SALAD (CVPR24) |

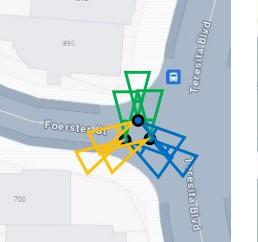

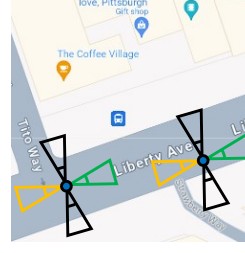

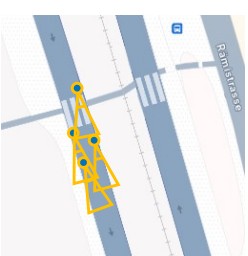

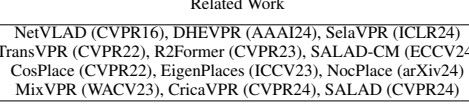

GSV-Cities  SF-XL  Pittsburgh  MSLS

Figure 3: **Schematic diagram of collecting VPR data.** VPR images with the same label are drawn using the same color in each minimal grid map. Although orange triangles appear in all four subgraphs, they represent different labels in each. The black triangles indicate that images have not been assigned labels.

following steps: 1) Perform grid partitioning as in SF-XL but without the heading label. 2) Use the $0°, 90°, 180°, 270°$ slices of panoramic images as subclass queries to search for similar training images in each grid partition using local feature matching (Sarlin et al., 2020; Lindenberger et al., 2023).

**MSLS (M)** originates from bicycle-mounted cameras, which typically capture images in a single direction. Therefore, we can use the grid partitioning method for classification without considering orientation information. It is worth noting that there is no need to set $\alpha$ and L in the step of aligning P and M.

## 4 EXPERIMENTS

In this section, we present a comprehensive set of experiments designed to rigorously evaluate the effectiveness of our proposed contributions. First, we outline the implementation details, including descriptions of the training and test sets, architectures, training configurations, and evaluation metrics. Following this, we provide a detailed comparative analysis of performance and a univariate analysis of each contribution.

### 4.1 IMPLEMENTATION DETAILS

Our training and evaluation code was built upon publicly available repositories, including DINOv2, MixVPR, NetVLAD, GeM, CosPlace, SelaVPR, SALAD, and Deep Visual Geo-localization Benchmark.

**Training sets.** Detailed descriptions of the GPMS dataset can be found in Tab. 1, Sec. 3.3, and the supplementary materials. Here, we emphasize the **fairness** of our experimental design: (1) In Tab. 3 and 4, we employed GPMS, which differs from the training sets used by other methods. This divergence reflects the contribution of SLA, and previous methods have also used varying training sets. (2) In Tab. 5 - 11, all experiments within each table are conducted on the **same** training set. For example, G$^2$M and NVL-FT$^2$ were trained on GSV-Cities, while SALAD was trained on GPMS in Tab. 10.

**Test sets.** We conducted our experiments across the **12** test sets, each representing distinct real-world challenges for VPR systems. A summary of these test sets is provided in Tab. 2. (1) The Pitts-30k test set (Arandjelovic et al., 2016), extracted from GSV, features significant viewpoint changes. As a

Table 2: **Overview of test sets.** These datasets have huge variations in size and domain shifts.

| Dataset Name | Pitts-30k test | Tokyo 24/7 | MSLS val | MSLS challenge | Nordland | Amster Time | SPED | SF-XL test-v1 | SF-XL test-v2 | SF-XL occlusion | SF-XL night | SVOX |
|---|---|---|---|---|---|---|---|---|---|---|---|---|
| # queries | 6.8k | 315 | 740 | 27,092 | 27592 | 1231 | 607 | 1000 | 598 | 466 | 76 | 4536 |
| # database | 10k | 76k | 18.9k | 38,770 | 27592 | 1231 | 607 | 2.8M | 2.8M | 2.8M | 2.8M | 17k |
| Scenery | urban | urban | various | various | country | urban | various | urban | urban | urban | urban | various |
| Domain | none | day/night | day/night | day/night | season | long-term | long-term | viewpoint | viewpoint | occlusion | day/night | weather |

subset of the larger Pitts-250k test set, Pitts-30k tends to yield lower performance metrics, suggesting greater difficulty and offering more room for improvement. (2) Tokyo 24/7 (Torii et al., 2017), consisting of database images sourced from GSV and query images captured by mobile devices, includes significant variations in lighting and perspective. (3) The MSLS dataset (Warburg et al., 2020) is collected from driving recorders worldwide, presenting numerous challenging scenarios, such as weather and seasonal variations, day/night transitions, and complex road conditions. This dataset includes two test subsets: val and challenge. The ground truth for the challenge subset is unavailable, and VPR performance is evaluated using an online ranking system. More details are presented in supplementary materials.

**Architecture.** We selected DINOv2 as a pre-trained model for two reasons: (1) It provides a fair benchmark for comparing SuperPlace with recent methods, and (2) Even though DINOv2 was released over a year ago, it remains the most effective pre-trained model available.

**Training configurations.** The experiments were conducted on a server with 8 NVIDIA 4090 GPUs. Instead of the Parameter-Efficient Fine-Tuning (PEFT) approach used in SelaVPR (Lu et al., 2024b) and CricaVPR (Lu et al., 2024a), we adopted the fine-tuning of the last four layers (FT4) as used in SALAD (Izquierdo & Civera, 2024a). Specifically, BoQ fine-tuned only the last two layers of DINOv2 with a warm-up step. Although BoQ only adjusted the last two layers of DINOv2, it introduced many parameters and a warm-up step, increasing the training time and the number of training parameters, making it less efficient compared to FT4.

Unless otherwise specified, all experimental parameters followed these settings: (1) DINOv2-B (Base) was used as the pre-trained model. (2) The GCA module in $G^2M$ was set with a rank of 64, used GELU as the activation function, and had an output feature dimension of 768. (3) NV and NVL utilized 64 cluster centers, with NVL having an output feature dimension of 8192. (4) SuperPlace was trained using the Adam optimizer with the learning rate set to $6 \times 10^{-5}$ and the batch size set to 64. (5) In Tab. 3 and 4, SuperPlace was trained with the resolution of $322 \times 322$ (for best performance). In Tab. 5 - 11, the training resolution was set to $224 \times 224$ (for fast experiments).

**Evaluation metrics.** We followed the same evaluation metric as in existing literature (Arandjelovic et al., 2016; Berton et al., 2022b), where the recall@K is measured. Recall@K is the percentage of query images for which at least one of the top-K predicted reference images falls within a pre-defined threshold distance. Following common evaluation procedures, we set the threshold to 25 meters for the test sets with GPS label, $\pm 10$ frames for Nordland (Sünderhauf et al., 2013), and the corresponding matching image for SPED (Chen et al., 2017) and AmsterTime (Yildiz et al., 2022).

## 4.2 COMPARISON WITH STATE-OF-THE-ART METHODS

We conducted an extensive set of experiments to thoroughly evaluate the soundness of SuperPlace, comparing it against a wide range of methods. As shown in Tab. 3, this includes nine 1-stage retrieval methods: NetVLAD (Arandjelovic et al., 2016), SFRS (Ge et al., 2020), CosPlace (Berton et al., 2022a), MixVPR (Ali-bey et al., 2023), EigenPlaces (Berton et al., 2023), CricaVPR (Lu et al., 2024a), SALAD (Izquierdo & Civera, 2024a), BoQ (Ali-bey et al., 2024), and SALAD-CM (Izquierdo & Civera, 2024b), and five 2-stage re-ranking methods: Patch-NetVLAD (Hausler et al., 2021), TransVPR (Wang et al., 2022), R2Former (Zhu et al., 2023), SelaVPR (Lu et al., 2024b), EffoVPR (Tzachor et al., 2024). Previous studies typically avoided comparing 1-stage with 2-stage methods, as the former were generally considered inferior under equivalent conditions. However, our findings demonstrate that SuperPlace can outperform the 2-stage methods.

The key findings from our comprehensive experiments are summarized as follows:

Table 3: **Comparison to state-of-the-art methods on benchmark datasets.** The best is highlighted in **bold** and the second is underlined . † These methods were tested using two models trained separately on MSLS and Pittsbugh-30k. ‡ The results reported by CricaVPR use multiple (16) query images, so we additionally report the results of a single query image.

| | Method | Pre-trained model | Training set | Feat. dim. | MSLS-challenge | | | Pitts-30k-test | | | Tokyo-24/7 | | | MSLS-val | | |
|---|---|---|---|---|---|---|---|---|---|---|---|---|---|---|---|---|
| | | | | | R@1 | R@5 | R@10 | R@1 | R@5 | R@10 | R@1 | R@5 | R@10 | R@1 | R@5 | R@10 |
| 2-stage | Patch-NV† | VGG-16 | M, P | 2826×4096 | 48.1 | 57.6 | 60.5 | 88.7 | 94.5 | 95.9 | 86.0 | 88.6 | 90.5 | 79.5 | 86.2 | 87.7 |
| | TransVPR† | ViT | M, P | 1200×256 | 63.9 | 74.0 | 77.5 | 89.0 | 94.9 | 96.2 | 79.0 | 82.2 | 85.1 | 86.8 | 91.2 | 92.4 |
| | R2Former† | ViT | M, P | 500×131 | 73.0 | 85.9 | 88.8 | 91.1 | 95.2 | 96.3 | 88.6 | 91.4 | 91.7 | 89.7 | 95.0 | 96.2 |
| | SelaVPR† | DINOv2-L | M, P | 61×61×128 | 73.5 | 87.5 | 90.6 | 92.8 | 96.8 | 97.7 | 94.0 | 96.8 | 97.5 | 90.8 | 96.4 | 97.2 |
| | EffoVPR | DINOv2-L | S | 649×1024 | 79.0 | 89.0 | 91.6 | 93.9 | 97.4 | 98.5 | **98.7** | **98.7** | **98.7** | 92.8 | 97.2 | 97.4 |
| 1-stage | NetVLAD | VGG-16 | P | 32768 | 35.1 | 47.4 | 51.7 | 81.9 | 91.2 | 93.7 | 60.6 | 68.9 | 74.6 | 53.1 | 66.5 | 71.1 |
| | SFRS | VGG-16 | P | 4096 | 41.6 | 52.0 | 56.3 | 89.4 | 94.7 | 95.9 | 81.0 | 88.3 | 92.4 | 69.2 | 80.3 | 83.1 |
| | CosPlace | ResNet-50 | S | 2048 | 66.9 | 77.1 | 80.6 | 90.9 | 95.7 | 96.7 | 87.3 | 94.0 | 95.6 | 87.2 | 94.1 | 94.9 |
| | MixVPR | ResNet-50 | G | 4096 | 64.0 | 75.9 | 80.6 | 91.5 | 95.5 | 96.3 | 85.1 | 91.7 | 94.3 | 88.0 | 92.7 | 94.6 |
| | EigenPlaces | ResNet-50 | S | 2048 | 67.4 | 77.1 | 81.7 | 92.5 | 96.8 | 97.6 | 93.0 | 96.2 | 97.5 | 89.1 | 93.8 | 95.0 |
| | CricaVPR-16 ‡ | DINOv2-B | G | 4096 | 69.0 | 82.1 | 85.7 | 94.9‡ | 97.3 | 98.2 | 93.0 | 97.5 | 98.1 | 90.0 | 95.4 | 96.4 |
| | CricaVPR-1 ‡ | DINOv2-B | G | 4096 | 66.9 | 79.3 | 82.3 | 91.6 | 95.7 | 96.9 | 89.5 | 94.6 | 96.2 | 88.5 | 95.1 | 95.7 |
| | SALAD | DINOv2-B | G | 8448 | 75.0 | 88.8 | 91.3 | 92.4 | 96.3 | 97.4 | 94.6 | 97.5 | 97.8 | 92.2 | 96.2 | 97.0 |
| | BoQ | DINOv2-B | G | 12288 | 79.0 | 90.3 | 92.0 | 93.7 | 97.1 | 97.9 | 98.1 | 98.1 | **98.7** | 93.8 | 96.8 | 97.0 |
| | SALAD-CM | DINOv2-B | GM | 8448 | 82.7 | 91.2 | 92.7 | 92.6 | 96.8 | 97.8 | 96.8 | 97.5 | 97.8 | 94.2 | 97.2 | 97.4 |
| | *G²M* | DINOv2-B | G | **768** | 72.5 | 86.0 | 88.7 | 92.2 | 96.1 | 97.4 | 92.7 | 96.8 | 97.8 | 91.0 | 96.1 | 96.9 |
| | *NVL-FT²* | DINOv2-B | G | 8192 | 76.0 | 87.2 | 90.2 | 93.1 | 96.1 | 96.9 | 97.8 | 98.7 | 99.1 | 93.1 | 96.4 | 96.8 |
| | *SP-G²M* | DINOv2-B | GPMS | **768** | 79.1 | 90.1 | 92.0 | 92.2 | 96.2 | 97.3 | 94.3 | 97.6 | 97.8 | 93.2 | 96.8 | 97.4 |
| | *SP-NVL-FT²* | DINOv2-B | GPMS | 8192 | 80.4 | 92.5 | 93.6 | 93.7 | 97.4 | 98.2 | 96.8 | 98.4 | **98.7** | 94.3 | 97.2 | 97.7 |
| | *SP-NVL-FT²* | DINOv2-L | GPMS | 8192 | **84.8** | **93.1** | **94.2** | **94.1** | **97.8** | **98.5** | 97.1 | 98.4 | **98.7** | **94.5** | **97.8** | **98.1** |

Table 4: **Comparison (R@1) to SOTA methods on more challenging datasets.**

| Method | Pre-trained model | Feat. dim. | Nordland | Amster time | SPED | SF-XL test-v1 | SF-XL test-v2 | SF-XL occlusion | SF-XL night | SVOX |
|---|---|---|---|---|---|---|---|---|---|---|
| SelaVPR | DINOv2-L | / | 72.3 | 55.2 | 88.6 | 74.9 | 89.3 | 35.5 | 38.4 | 97.2 |
| SALAD | DINOv2-B | 8448 | 90.0 | 58.8 | 92.1 | 88.6 | **94.8** | 51.3 | 46.6 | 98.2 |
| BoQ | DINOv2-B | 12288 | 90.6 | **63.0** | **92.5** | - | - | - | - | **99.0** |
| *SP-G²M* | DINOv2-B | **768** | 88.0 | 54.4 | 87.3 | 84.0 | 92.3 | 43.4 | 38.2 | 98.1 |
| *SP-NVL-FT²* | DINOv2-B | 8192 | **91.4** | 62.3 | 87.5 | **90.9** | 94.1 | **59.2** | 45.3 | 98.6 |

1) Inspired by SelaVPR (ICLR'24), we trained SuperPlace using DINOv2-L as the pre-trained model. Although SelaVPR employs re-ranking and the MSLS training set, its Recall@1 is 11.3% lower than that of SP-NVL-FT² on the MSLS-challenge dataset.

2) CricaVPR has a query leakage issue, which disqualifies it from fair comparison on the Pitts-30k test set. Beyond this, CricaVPR's overall Recall@K performance is inferior to both SALAD and BoQ. Although BoQ outperforms SALAD, its higher feature dimensions should be considered.

3) The overall recall@K of SP-NVL-FT²(B) and BoQ is evenly matched, with NVL-FT² benefiting from the training set and BoQ from its larger feature dimensions and higher number of parameters.

4) SP-G²M achieves competitive results with significantly smaller dimensions than other methods, making it suitable for real-time applications in large-scale environments.

5) High-dimensional representations and re-ranking offer advantages in handling day-night variations, so SP-NVL-FT² performs slightly worse than EffoVPR and BoQ on Tokyo 24/7.

6) When tested on large datasets, the limitations of feature dimensions become apparent. For instance, our platform could not evaluate BoQ's performance on the SF-XL dataset due to its large feature dimensions.

We provide another perspective of the analysis in the supplementary material, focusing on different configurations (pre-trained models and datasets).

Table 5: **Ablation of the GPMS dataset.**

| | G | P | M | S | Pitts-30k-test | | MSLS-val | | SF-XL-val | |
|---|---|---|---|---|---|---|---|---|---|---|
| | | | | | R@1 | R@5 | R@1 | R@5 | R@1 | R@5 |
| G2M | ✓ | | | | 92.6 | 96.8 | 90.4 | 95.9 | 91.2 | 95.8 |
| | ✓ | ✓ | | | **93.1** | **96.9** | 90.8 | **96.6** | 91.8 | 96.1 |
| | ✓ | | ✓ | | 92.3 | **96.9** | **91.5** | **96.6** | 92.2 | 96.5 |
| | ✓ | | | ✓ | 92.2 | 96.6 | 89.6 | 96.1 | **92.3** | 96.7 |

Table 6: **Comparison to CliqueMining.**

| Method | Training Set | Pitts-30k-test | | MSLS-val | | MSLS-challenge | |
|---|---|---|---|---|---|---|---|
| | | R@1 | R@5 | R@1 | R@5 | R@1 | R@5 |
| SALAD-CM | G+M | 92.6 | 96.8 | 94.2 | 97.2 | **82.7** | 91.2 |
| SALAD-SLA | | **93.0** | **97.1** | **94.3** | **97.8** | 82.1 | **93.5** |

Table 7: Comparison of different implements of DINOv2-GeM and channel attention modules.

| | Ablated Versions | Pitts-30k-test R@1 | Pitts-30k-test R@5 | Tokyo-24/7 R@1 | Tokyo-24/7 R@5 | MSLS-val R@1 | MSLS-val R@5 |
|---|---|---|---|---|---|---|---|
| GSV-Cities | Frozen-DINOv2-GeM | 74.8 | 90.1 | 49.8 | 67.0 | 45.4 | 60.7 |
| | Adapt-GeM (CricaVPR) | 87.1 | 94.0 | 70.2 | 85.4 | 78.4 | 87.8 |
| | FT4-GeM (SALAD) | - | - | - | - | 85.4 | 93.9 |
| | FT4-GeM (Our impl.) | 91.9 | 96.6 | 94.3 | 97.8 | 90.3 | 95.4 |
| | GeM + SE | 91.5 | 96.0 | 93.0 | 97.5 | 90.5 | 95.7 |
| | GeM + CBA | 91.6 | 96.2 | 92.4 | 98.0 | 90.1 | 95.4 |
| | GeM + GCA ($G^2M$) | 92.6 | 96.8 | 94.0 | 98.1 | 90.4 | 95.9 |

Table 8: Comparison of different ranks and activate functions for $G^2M$.

| | Ablated versions | Rank | Pitts-30k-test R@1 | Pitts-30k-test R@5 | Tokyo-24/7 R@1 | Tokyo-24/7 R@5 | MSLS-val R@1 | MSLS-val R@5 |
|---|---|---|---|---|---|---|---|---|
| GSV-Cities | GeM | / | 91.9 | 96.6 | 94.3 | 97.8 | 90.3 | 95.4 |
| | $G^2M$ (GELU) | 3 | 91.9 | 96.4 | 95.2 | 96.5 | 90.9 | 95.8 |
| | | 32 | 91.7 | 96.6 | 93.0 | 97.5 | 90.5 | 94.9 |
| | | 64 | 92.6 | 96.8 | 94.0 | 98.1 | 90.4 | 95.9 |
| | | 128 | 91.4 | 96.4 | 93.7 | 97.8 | 90.9 | 95.5 |
| | $G^2M$ (ReLU) | 64 | 92.5 | 96.8 | 94.9 | 97.5 | 90.1 | 95.4 |

## 4.3 Univariate experiment of Supervised Label Alignment

**Contribution of each component of GPMS.** We conducted an ablation experiment to evaluate the contribution of each subset of the GPMS dataset, as shown in Tab. 5. First, SP-$G^2M$ was trained on the G dataset and then fine-tuned for one epoch each on the P, M, and S datasets, respectively. The bolded results align approximately along the diagonal in Tab. 5, indicating that each dataset contributes most effectively to its corresponding test set. This indicates that the datasets do not completely encompass each other's characteristics.

**Comparison with another alignment method.** We compared our method, SLA, with another alignment approach (Izquierdo & Civera, 2024b) published in a forthcoming ECCV that employed CliqueMining (CM) to mix GSV-Cities and MSLS datasets. Despite the two works being nearly concurrent, we ensured a fair comparison to highlight the superiority of our approach. As shown in Tab. 6, SLA outperforms CM using the DINOv2-SALAD framework.

## 4.4 Univariate experiment of the improved GeM

**Ablation and Comparison for $G^2M$.** We only used the feature aggregator as a variable to conduct experiments to verify the effectiveness of $G^2M$, as shown in Tab. 7. First, we explored using different fine-tuning methods for DINOv2-GeM: freezing, using adaptors (a PEFT method), and fine-tuning the last four layers (FT4). In particular, FT4-GeM has two versions: our implementation and SALAD's implementation (Izquierdo & Civera, 2024a). We found that DINOv2-GeM can achieve state-of-the-art performance, but recent works have not reproduced this effectiveness (Lu et al., 2024a; Izquierdo & Civera, 2024a). Then, we added three modules to GeM: SE, CBA, and our proposed GCA. Here, we set the rank $r = 64$ recommended in the SE (Hu et al., 2018) and CBA (Woo et al., 2018), consistent with the GCA rank we selected in Tab. 8. The GCA module is better than the other two.

**Design details of $G^2M$.** As shown in Tab. 8, we adjusted the rank and activation function of GCA to improve the design of GCA. Since the distributions of GSV-Cities and Pitts-30k were closely related, we mainly selected parameters based on the results of Pitts-30k.

## 4.5 Univariate experiment of the improved NetVLAD

**Design details of NVL-FT[2].** As shown in Tab. 9, we conducted detailed design experiments and training analyses for NVL. NVL-FT[2] more closely approximates the performance of NV, outperforming one-shot NVL, twice-fine-tuned NV-MLP, and NV-PCA. We also found that incorporating a CLS Token into NVL did not improve performance. Observing the training time and number of training parameters, we found that although the steps in FT[2] are more complex, the overall efficiency improves.

**Comparison with SALAD.** We only used the aggregator as a variable to conduct comparative experiments with SALAD. It is important to note that Izquierdo & Civera (2024a) conducted comparative experiments with NetVLAD but did not use the recommended parameters from Arandjelovic et al. (2016). As shown in Tab. 10, NetVLAD is better than SALAD but has the disadvantage of too high a dimension. NVL-FT[2] overcomes this limitation and surpasses SALAD in performance. Izquierdo & Civera (2024a) also claimed that SALAD could be scaled to ultra-low dimensions (544-dim) while maintaining good performance. We conclude that $G^2M$ offers the best performance compared to low-dimensional methods.

Table 9: **Comparison of variant versions of NV.** Training time (min) was measured on four 4090 GPUs, while inference time (ms) of the aggregation layer was measured on a single 4090 GPU.

| | Aggre. | CLS | FT[2] | Feat. dim. | Pitts-30k-test R@1 | Pitts-30k-test R@5 | MSLS-val R@1 | MSLS-val R@5 | Train. time | Trainable param. (M) | Infer. time |
|---|---|---|---|---|---|---|---|---|---|---|---|
| GPMS | NV | / | / | 49152 | 93.5 | 97.4 | 94.6 | 97.6 | 22 | 27.139 | 4.39 |
| | NV-PCA | / | / | 8192 | 93.2 | 97.3 | 94.2 | 97.3 | / | / | 11.1 |
| | NV-MLP | | ✓ | 8192 | 93.0 | 97.2 | 93.8 | 97.2 | 22 + 28 | 0.438 | 4.66 |
| | NVL | | | 8192 | 93.0 | 97.1 | 93.8 | 97.3 | 34 | 27.231 | 4.49 |
| | NVL | ✓ | | 8448 | 93.1 | 97.2 | 92.8 | 97.2 | 51 | 27.418 | 4.60 |
| | NVL | ✓ | ✓ | 8448 | 91.5 | 95.8 | 91.7 | 96.2 | 22 + 28 | 0.281 | 4.60 |
| | **NVL-FT[2]** | | ✓ | 8192 | **93.1** | **97.4** | **94.6** | **97.8** | 22 + 14 | **0.094** | 4.49 |

Table 10: **Comparison to SALAD.**

| | Method | Feat. dim | Pitts-30k-test R@1 | Pitts-30k-test R@5 | MSLS-val R@1 | MSLS-val R@5 | SF-XL-val R@1 | SF-XL-val R@5 |
|---|---|---|---|---|---|---|---|---|
| GPMS | NetVLAD | 49152 | **93.5** | **97.4** | 94.6 | 97.6 | **96.3** | **98.2** |
| | SALAD | 8448 | 92.8 | 96.9 | **94.7** | 97.4 | 94.8 | 98.3 |
| | **NVL-FT[2]** | 8192 | 93.1 | **97.4** | 94.6 | **97.8** | 95.5 | 98.0 |
| | GeM | 768 | **92.4** | **96.8** | 91.5 | 96.4 | 92.6 | 97.0 |
| | **G²M** | 768 | 92.0 | 96.6 | **92.4** | **96.8** | **93.2** | **97.4** |
| | SALAD | 544 | 91.3 | 96.6 | 92.3 | **96.8** | 92.9 | 97.1 |

Table 11: **Comparison to BoQ.** [†] The fine-tuning with warm-up is used.

| | Method | Param. (M) | Infer. time (ms) | Feat. dim. | Pitts-30k-test R@1 | Pitts-30k-test R@5 | MSLS-val R@1 | MSLS-val R@5 |
|---|---|---|---|---|---|---|---|---|
| GSv-Cities | BoQ[†] | 8.63 | 2.53 | 12288 | **93.7** | **97.1** | **93.8** | **96.8** |
| | SALAD | 1.41 | 1.45 | 8448 | 92.4 | 96.3 | 92.2 | 96.2 |
| | NVL-FT[2] | 0.197 | 4.49 | 8192 | 93.0 | 96.7 | 93.0 | 96.5 |
| | NVL-FT[2] [†] | 0.197 | 4.49 | 8192 | 93.4 | 97.0 | 93.1 | 96.6 |
| | G²M | 0.69 | 0.41 | 768 | 92.6 | 96.8 | 90.4 | 95.9 |

**Comparison with BoQ.** We only used the aggregator as a variable to conduct comparative experiments with BoQ, as shown in Tab. 11. BoQ used a warm-up training technique suitable for training large parameter structures at the expense of longer training time. In addition, the training resolution of BoQ is $280 \times 280$. We applied this technique and resolution to NV and observed a slight performance improvement. Considering only Recall@K, BoQ is slightly better than NVL-FT[2]. However, considering BoQ's complex training techniques, extended training time, large parameters, and high feature dimensions, NVL-FT[2] is a more efficient solution.

## 5 CONCLUSION

This paper presents SuperPlace, a novel VPR system that integrates classical aggregation methods and modern foundation models to achieve state-of-the-art performance. Specifically, we propose three contributions: 1) a supervised label alignment method that combines grid partitioning and local feature matching, allowing models to be trained on diverse VPR datasets within a unified framework akin to the design principles of foundation models. 2) G²M, a compact feature aggregation with two GeM layers, in which one GeM learns the principal components of feature maps along the channel direction and calibrates the other GeM output. 3) the secondary fine-tuning (FT²) strategy for NetVLAD-Linear. NetVLAD first learns feature vectors in a high-dimensional space and then compresses them into a low-dimensional space by a single linear layer. Extensive experiments have validated the effectiveness of SuperPlace, with SuperPlace-G²M achieving high performance with minimal dimensions and SuperPlace-NVL-FT² dominating the MSLS Challenge leaderboard. These results highlight the strength of revisiting and refining classical methods in the era of visual foundation models. In the future, we will further explore developing interpretable and open-world VPR systems.

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

# A  APPENDIX

## A.1  THE SNAPSHOT OF MSLS LEADERBOARD

The MSLS place recognition challenge [1] is an authoritative competition for VPR with over 100 participants. Fig. 4 shows a snapshot of the MSLS challenge leaderboard at the time of submission. The proposed method (named "SuperPlace" due to the double-blind review policy) ranks first.

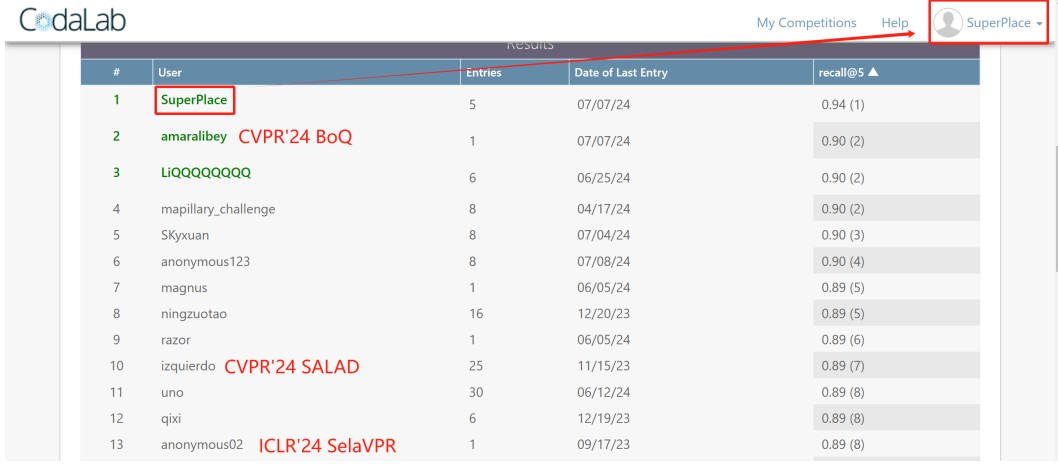

Figure 4: **A snapshot of MSLS leaderboard.** The upper-right corner of the screenshot indicates our username. By consulting the supplementary materials of SelaVPR (Lu et al., 2024b), we confirm that 'anonymous02' corresponds to SelaVPR.

## A.2  COMPARISON OF DIFFERENT DIMENSIONS

We further explored the performance of $G^2M$ and NVL-FT$^2$ in different dimensions, with other parameters consistent with the SP in Tab. 3. As shown in Fig. 5, NVL-FT$^2$ shows a significant performance improvement as the dimension increases, but the growth is relatively weak after exceeding 8000 dimensions. The performance trend of $G^2M$ is less consistent, and we recommend maintaining a feature dimension aligned with the number of channels in the extracted feature map.

## A.3  COMPARISON WITH STATE-OF-THE-ART METHODS

---

[1]https://codalab.lisn.upsaclay.fr/competitions/865

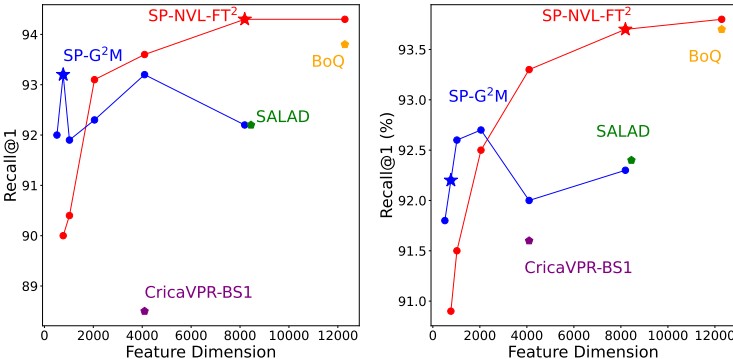

Figure 5: The Recall@1 and descriptor dimensionality comparison of different methods on MSLS-Val (left) and Pitts-30k (right).

We provide further analysis in Tab. 3, focusing on different configurations (pre-trained models and datasets) and their contribution to the performance:

**1) Pre-trained Models:**

- VGG-16 and ResNet-50: Older backbones like VGG-16 (e.g., NetVLAD, SFRS) consistently underperform compared to modern pre-trained vision transformers (e.g., DINOv2). For instance, NetVLAD achieves only 35.1% R@1 on MSLS-challenge, whereas modern methods using DINOv2 (e.g., SALAD, SP-NVL-FT$^2$) exceed 45%. This underscores the importance of leveraging advanced pre-trained models for feature extraction.

- DINOv2 Variants: Methods using DINOv2-L (e.g., SP-NVL-FT$^2$) outperform those using the smaller DINOv2-B due to the former's ability to extract more robust representations, especially on challenging datasets like MSLS.

**2) Feature Dimensions:**

- Low-Dimensional Efficiency: SP-G$^2$M achieves competitive results with a feature dimension of only 768, demonstrating its suitability for memory-constrained real-time applications.

- High-Dimensional Representations: Methods like BoQ and SALAD-CM leverage larger dimensions (e.g., 12,288 for BoQ), improving performance in scenarios with significant appearance variations (e.g., day-night changes). However, these approaches face limitations on large-scale datasets due to higher computational and memory requirements, as noted in the inability to evaluate BoQ on SF-XL.

**3) Training sets:**

- GSV-Cities is widely used by multiple methods due to its diversity of cities and training efficiency. Models trained on this dataset are less likely to exhibit over-optimization or suboptimal performance specific to a particular test set.

- Models trained on the SF-XL dataset perform well on the Pitts-30k test set and the Tokyo-247 dataset but are relatively less optimal on the MSLS-val and MSLS-challenge dataset. This phenomenon is largely due to the fact that the former datasets originate from Google Street View services, sharing certain common characteristics, while the latter comes from a crowdsourced dataset.

- The MSLS training set and the Pittsburgh training set have seen less frequent use recently. However, our method's results demonstrate their substantial value, particularly in enhancing a model's generalization capability in challenging scenarios.

Table 12: **The proposed aggregations work with ResNet-50.** † These results are reported in Ali-bey et al. (2022).

| | Aggregations | Training set | Feat. dim. | Pitts-30k-test R@1 | R@5 | MSLS-val R@1 | R@5 | SF-XL-val R@1 | R@5 |
|---|---|---|---|---|---|---|---|---|---|
| ResNet-50 | GeM† | G | 2048 | - | - | 76.5 | 85.7 | - | - |
| | GeM | G | 1024 | 89.8 | 94.8 | 84.5 | 90.5 | 87.7 | 93.3 |
| | GeM | GPMS | 1024 | 90.4 | 95.3 | 87.6 | 92.8 | 90.1 | 94.8 |
| | G²M | GPMS | 1024 | 90.4 | 94.8 | 88.8 | 93.2 | 88.3 | 93.8 |
| | NetVLAD | G | 65536 | 89.9 | 95.0 | 78.9 | 87.7 | 84.9 | 92 |
| | NetVLAD | GPMS | 65536 | 90.4 | 95.3 | 89.2 | 93.8 | 88.9 | 94.1 |
| | NVL | GPMS | 8192 | 89.9 | 94.9 | 87.6 | 93.8 | 88.2 | 93.7 |
| | NVL-FT²† | GPMS | 8192 | 90.6 | 95.2 | 88 | 93.3 | 88.2 | 93.8 |

Table 13: **The proposed aggregations work with CLIP**

| | Aggregations | Training set | Feat. dim. | Pitts-30k-test R@1 | R@5 | MSLS-val R@1 | R@5 | SF-XL-val R@1 | R@5 |
|---|---|---|---|---|---|---|---|---|---|
| CLIP | GeM | G | 1024 | 86.8 | 94.7 | 79.2 | 88.8 | 81.1 | 89.4 |
| | GeM | GPMS | 1024 | 88.0 | 94.8 | 85.3 | 93.1 | 83 | 91.1 |
| | G²M | GPMS | 1024 | 89.1 | 95 | 86.1 | 93.5 | 85.4 | 92.7 |
| | NetVLAD | G | 49152 | 89.3 | 95.4 | 82.8 | 91.1 | 83.7 | 90.9 |
| | NetVLAD | GPMS | 49152 | 90.2 | 95.7 | 88.1 | 93.7 | 86.7 | 93.7 |
| | NVL | GPMS | 8192 | 89.6 | 95.4 | 86.4 | 93.4 | 84.8 | 92.7 |
| | NVL-FT² | GPMS | 8192 | 89.9 | 95.5 | 87.0 | 93.4 | 86.1 | 93.0 |

## A.4 PERFORMANCE ON OTHER FOUNDATION MODELS

We further explored the performance of G²M and NVL-FT² on other foundation architectures or models. VGG and ResNet are both convolutional architectures, and the theoretical structures of DINOv2 and ViT are identical, differing primarily in their improved model parameters. Therefore, we believe it is sufficient to conduct generalization experiments for architectures using ResNet-50. For validating the generalization capability of the foundation models, we selected the visual encoder of OpenAI's CLIP model, as it is widely recognized by researchers. As shown in Tab. 12 and 13, our method achieves the same conclusions on ResNet-50 and CLIP as reported in the main text. Notably, the experimental results of DINOv2 remain the best.

## A.5 DATASET DETAILS

**Pittsburgh-250k** (Arandjelovic et al., 2016) is collected from Google Street View and provides 24 images with different viewpoints at each place. The images in this dataset have large viewpoint variations and moderate condition variations.

**Tokyo24/7** (Torii et al., 2017) includes 75,984 database images and 315 query images captured from urban scenes. The query images are selected from 1,125 images taken at 125 distinct places with three different viewpoints and at three different times of day. Significant viewpoint and condition changes (e.g., day-night transitions) are present.

**Mapillary Street-Level Sequences (MSLS)** (Warburg et al., 2020) is a large-scale VPR dataset containing over 1.6 million images labeled with GPS coordinates and compass angles, captured from 30 cities in urban, suburban, and natural scenes over seven years. It covers various challenging visual changes due to illumination, weather, season, viewpoint, and dynamic objects. It includes subsets of training, public validation (MSLS-val), and withheld test (MSLS-challenge).

**Nordland** (Sünderhauf et al., 2013) primarily consists of suburban and natural place images captured from the same viewpoint in the front of a train across four seasons, which results in severe condition changes (e.g., seasons and lighting) but no viewpoint variations. Its ground truth is provided by the frame-level correspondence. Following previous work (Sünderhauf et al., 2013; Wang et al., 2022), we use the dataset partition first presented in (Sünderhauf et al., 2013) for our experiments.

**AmsterTime** (Yildiz et al., 2022) is a collection of over one thousand pairs of query-reference images of Amsterdam. For each pair, the query is a grayscale historical image, and its reference is a modern-day photo that represents the same place, as confirmed by human experts. The pairs exhibit multiple domain shifts, including changes in viewpoint, long-term temporal variations, modality differences (RGB vs. grayscale), and different camera systems. Despite its relatively small scale, AmsterTime is one of the most challenging datasets available.

**SPED** (Chen et al., 2017) comprises low-quality, high-scene-depth images taken from CCTV cameras around the globe. The images in this dataset show various condition variations, such as lighting, weather, and seasonal changes. This dataset covers various outdoor scenes, including forest landscapes, country roads, and urban environments.

**SF-XL** (Berton et al., 2022a) is a huge dataset covering San Francisco with over 41M images. Its test set covers the same with a less dense set of 2.8M images. Two sets of queries are used: the first

(test v1) is a challenging set of 1000 images from Flickr, with multiple challenges like night images and photos from the sidewalk. Test v2 uses the same set of queries from San Francisco Landmark.

**SVOX** (Berton et al., 2021b) is a cross-domain dataset built from cross-domain VPR that evaluates multiple weather conditions. It spans the city of Oxford, with a large (single domain) database from GSV images: the queries are instead from the Oxford RobotCar dataset (Maddern et al., 2017), providing several weather conditions, such as overcast, rainy, sunny, snowy, and night domains.

