# OpenReview forum: "The Renaissance of Classic Feature Aggregations for Visual Place Recognition in the Era of Foundation Models"
_ICLR.cc/2025/Conference — ICLR 2025 Conference Withdrawn Submission_

### Official Review · Reviewer_fhKJ · 2024-10-28

**Soundness:** 2
**Presentation:** 1
**Contribution:** 2
**Rating:** 5
**Confidence:** 3

**Summary:**

The work outlines three contributions to Visual Place Recognition (VPR). The first two are intended to improve the aggregation modules, such as NetVLAD (NVL-FT$^2$) and GeM (G$^2$M) of DINOv2 foundational features. At the same time, the third contribution addresses the discrepancy in existing VPR benchmarks and proposes a unified one that combines four datasets with a consistent labeling scheme.

**Strengths:**

The paper extensively and thoroughly presents the historical overview of the VPR field (introduction and related works sections).

**Weaknesses:**

1. The paper is hard to follow, and the main storyline of the work is missing. While some limitations are addressed (lack of consensus on the VPR model supervision, limitation of classical aggregation methods for FM features), the main challenge that unites the three contributions is unclear and is not explicitly stated. Experiments should be designed and presented according to the storyline, demonstrating the effectiveness of the proposed method in tackling a specific issue.

2. The experimental part contains a lot of information with a mixture of different settings (training data, pre-trained backbone models) and requires careful discussion and self-containing captions (Tables 3 - 8, 10, 11). Instead of a numbered summary in Section 4.2 it would be beneficial to discuss the differences between the performance of the proposed method and SOTA approaches, stressing on the advantages and disadvantages of the former on different configurations.

3. Novelty of the work is limited. Dimensionality compression of NetVLAD features has been proposed before [1, 2], while GCA components have been used as a simple baseline in feature fusion works [3].

4. The decision to quantize the training labels while evaluation still abides by the common practices of utilizing continuous coordinates appears to be a limitation. This concern should be addressed with further baselines (Table 6) and qualitative figures demonstrating the superiority of one convention over another.

[1] Gong et al. "Ghost-dil-NetVLAD: A Lightweight Neural Network for Visual Place Recognition", 2024

[2] Griange et al. "Design Space Exploration of Low-Bit Quantized Neural Networks for Visual Place Recognition", 2023

[3] Wu et al. "Asymmetric Feature Fusion for Image Retrieval", 2024

**Questions:**

1. I believe that the work in its current state requires a significant rewrite. There are two possible ways of approaching this task: (1) finding a common problem that can be effectively addressed with the new aggregation modules and multi-dataset training; (2) separating contributions into different works with more detailed discussions and extensive evaluations.
2. It is unclear what "SuperPlace" is and how it differs from DINOv2 or other foundational model (FM) backbone.
3. It is unclear if Supervised Label Alignment is performed during training or if it is a pre-processing algorithm.
4. The rationale behind combining four datasets, namely GSV-Cities, SF-XL, Pittsburgh-250k, and MSLS, is unclear. Does the main motivation lie solely in increasing the amount of training data or its diversity?
5. What does "rank" in GCA refer to (Section 4.4, Table 8)?
6. How do PCA components of feature maps look after training GCA in alignment with the location-based information (Figure 2)?
7. How are G$^2$M and NVL-FT$^2$ trained (loss function)?
8. It would be interesting to replace GeM with GCA

---

> ### Author Response · Authors · 2024-11-17
> **Response to Weakness**
>
> **Thank you for your thoughtful comments and suggestions. We hope the following clarifications address your concerns.**
>
> > *Weakness:* The paper is hard to follow, and the main storyline of the work is missing.
>
> > *Question:* I believe that the work in its current state requires a significant rewrite.
>
> **A:** A good story certainly aids in presenting research effectively. However, we know that some studies with limited performance gains may rely on storytelling to mislead readers unfamiliar with VPR. As highlighted by our title and various nuanced statements in the paper (L74, L206, L403, and L437), we aim to distinguish our work from such studies. We focus on presenting our method transparently, using straightforward language and solid experimental evidence. As noted at the end of the abstract, our supplementary materials include a ranking screenshot, the source code, and the original experimental records.
>
> > *Weakness:* it would be beneficial to discuss the differences between the performance of the proposed method and SOTA approaches, stressing on the advantages and disadvantages of the former on different configurations.
>
> **A:**
>
> > *Weakness:* Novelty of the work is limited. Dimensionality compression of NetVLAD features has been proposed before [1, 2], while GCA components have been used as a simple baseline in feature fusion works [3].
>
> **A:** We respectfully disagree with this assessment:
>
> 1. The dimensionality compression approach in [1] is **only** mentioned in a single sentence: The dimension reduction is performed using principal component analysis (PCA) with whitening followed by L2-normalization. This description is similar to what we discuss as prior methods in L208 of our paper. Moreover, the article does not specify the dimensionality of the aggregation layer.
>
> 2. The discussion in [2] is similarly sparse, with **only** this mention: The additional fully connected linear layer following the NetVLAD descriptor projects the representation to a size of 1024 and hence consumes a significant amount of memory. Interestingly, Figure 6 of [2] shows that NetVLAD performs worse than GeM, which contradicts our findings and thus supports our paper’s originality.
>
> 3. As for [3], Figure 4 illustrates a method where the input consists of n dx1 vectors, and the output is a single dx1 vector. In contrast, our method takes a single cx1 vector as input and outputs a cx1 vector, demonstrating a clear difference.
>
> > *Weakness:* The decision to quantize the training labels while evaluation still abides by the common practices of utilizing continuous coordinates appears to be a limitation.
>
> **A:** Sorry, I am struggling to fully understand your comment. Are you suggesting that using classification-based learning for metric problems might be inappropriate?
>
> **Updated on 23 Nov**
>
> **Apologies for not fully understanding your concern earlier. I encountered similar doubts two years ago, but after gaining clarity on the rationale, I mistakenly assumed this approach was a consensus in the VPR field.**
>
> 1. We appreciate your perspective. Firstly, we are not the first in the VPR field to use this approach [6, 7]. Both continuous coordinates and quantized labels are proxies for location similarity. However, quantized labels improve training efficiency by simplifying the learning task [6, 7].
>
> 2. We also experimented with using continuous coordinates for data alignment through triplet mining methods. (While the GitHub repository remains private due to double-blind review policies, the earliest implementation will be visible in the commit history once the repository is made public.) This approach, however, has limitations when applied to cropped images derived from panoramic sources. A notable issue arises when anchor and positive samples share the same GPS coordinates but do not share a common view. We explored two potential solutions:
>       - Dynamic Ranking Method [8]: Effective but significantly increases training time.
>       - Local Feature Matching for Verification: Handles view inconsistency but introduces complex loop logic.
>
> Ultimately, Supervised Label Alignment (SLA) proved to be a more concise and flexible solution. Notably, results from both approaches are consistent, but SLA’s streamlined logic makes it easier to extend and adapt.

---

> ### Author Response · Authors · 2024-11-17
> **Response to Question**
>
> > *Weakness:* This concern should be addressed with further baselines (Table 6) and qualitative figures demonstrating the superiority of one convention over another.
>
> **A:**
> 1. In Table 6, our method is intentionally tested on the same dataset to ensure a fair comparison, though our approach supports both Pittsburgh and SF-XL datasets, underscoring its superiority.
> 2. At the time of our ICLR submission, the SALAD-CM had not yet been officially published nor provided reproducible code with documentation. Our comparison with SALAD-CM is, therefore, an additional work rather than a core focus of our work.
> 3. Additionally, it's worth mentioning that two other VPR papers submitted to ICLR-2025 did not reference this study.
>
> >*Question:* It is unclear if Supervised Label Alignment is performed during training or if it is a pre-processing algorithm.
>
> **A:** Supervised Label Alignment (SLA) is a pre-processing algorithm.
>
> >*Question:* The rationale behind combining four datasets, namely GSV-Cities, SF-XL, Pittsburgh-250k, and MSLS, is unclear. Does the main motivation lie solely in increasing the amount of training data or its diversity?
>
> **A:** The rationale behind combining datasets such as GSV-Cities, SF-XL, Pittsburgh-250k, and MSLS goes beyond merely increasing the training data volume. Our motivation also emphasizes enhancing data diversity. This approach is inspired by scaling laws from the large language model (LLM) domain and aligns with recent trends in the research community [4][5]. While Ilya Sutskever has mentioned that LLM scaling laws might be reaching their limits, we believe that VPR research still has untapped potential for scaling benefits.
>
> >*Question:* What does "rank" in GCA refer to (Section 4.4, Table 8)?
>
> **A:** The "rank" in GCA refers to the internal representation dimension of the MLP. As mentioned in L194: This branch consists of a new GeM pooling layer, a low-**rank** MLP, a GELU activation, and a Sigmoid function.
>
> >*Question:* How are G2M and NVL-FT2 trained (loss function)?
>
> **A:** As stated in L160, we use multi-similarity loss (Wang et al., 2019) for training both G2M and NVL-FT2.
>
> >*Question:* It would be interesting to replace GeM with GCA.
>
> **A:** Thank you for your suggestion. We did consider replacing GeM with GCA in earlier experiments, but unfortunately, it did not yield improved results.
>
> ```
> [1] Gong et al. "Ghost-dil-NetVLAD: A Lightweight Neural Network for Visual Place Recognition", 2024
> [2] Griange et al. "Design Space Exploration of Low-Bit Quantized Neural Networks for Visual Place Recognition", 2023
> [3] Wu et al. "Asymmetric Feature Fusion for Image Retrieval", 2024
> [4] Wang, Shuzhe, et al. "Dust3r: Geometric 3d vision made easy." Proceedings of the IEEE/CVF Conference on Computer Vision and Pattern Recognition. 2024.
> [5] Yang, Lihe, et al. "Depth anything: Unleashing the power of large-scale unlabeled data." Proceedings of the IEEE/CVF Conference on Computer Vision and Pattern Recognition. 2024.
> [6] Berton, Gabriele, et al. "Rethinking visual geo-localization for large-scale applications." Proceedings of the IEEE/CVF Conference on Computer Vision and Pattern Recognition. 2022.
> [7] Ali-bey, Amar, Brahim Chaib-draa, and Philippe Giguere. "Gsv-cities: Toward appropriate supervised visual place recognition." Neurocomputing 513 (2022): 194-203.
> [8] Ge, Yixiao, et al. "Self-supervising fine-grained region similarities for large-scale image localization." Computer Vision–ECCV 2020: 16th European Conference, Glasgow, UK, August 23–28, 2020, Proceedings, Part IV 16. Springer International Publishing, 2020.
> ```
>
> We sincerely appreciate your constructive suggestions and believe that the additional experiments, analysis, and explanations significantly improve the quality of our submission.
>
> **We hope that this provides sufficient reasons to raise the score.**

---

> ### Author Response · Authors · 2024-11-21
>
> > *Question:* How do PCA components of feature maps look after training GCA in alignment with the location-based information (Figure 2)?
>
> **A:** We have visualized it according to your suggestion and added it to the supplementary materials. The file is named **GCA_feature_map.pdf**. As expected from our paper, location-relevant information (buildings) has a higher response, and location-irrelevant information (sky, dynamic objects) is given a lower response.
>
> **Dear Review fhKJ,**
>
> *Thank you again for the great efforts and valuable comments. We have carefully addressed the main concerns in detail. We hope you might find the response satisfactory. As the discussion phase is about to close, we are very much looking forward to hearing from you about any further feedback. We will very happy to clarify any further concers (if any).*
>
> **Best, Authors**

---

> ### Author Response · Authors · 2024-11-22
>
> **Thank you for your detailed feedback.**
>
> **We apologize for any inaccuracies in our statements.**
>
> **And we would like to kindly clarify that there seems to have been some misunderstanding regarding our intentions.**
>
> > *Weakness:* Concerning the incentive of having a good storyline, I would respectfully disagree that its sole purpose is to deceive the readers.
>
> **A:**
>
> 1. We did not claim that a "good storyline" is solely for deception. What we stated was: "*Some studies may rely on storytelling to mislead readers unfamiliar with VPR.*"
>
> 2. We have presented our own story (motivation), as you noted in earlier feedback. While we regret that it did not resonate with you, the issues we address (e.g., the lack of consensus on VPR supervision and limitations of classical aggregation methods) form a coherent and grounded narrative.
>
> 3. L74: However, through some tentative experiments, we found that classic methods from ten years ago are still competitive. This prompted us to improve these classic methods instead of following recent aggregations introduced in the past year.
>
> > *Weakness:* I agree with reviewer YFv3 about the confusion from presenting two competitive VPR methods in a single framework ...
>
> **A:**
>
> 1. The work you cited [2] and DVGL [4] also present experiments with multiple aggregation methods within a single framework.
>
> 2. We have already addressed YFv3’s concerns regarding this issue.
>
> 3. Splitting the contributions into separate papers might raise questions about insufficient novelty, whereas consolidating them highlights their combined value.
>
> 4. L74: However, through some tentative experiments, we found that classic methods from ten years ago are still competitive. (To put it bluntly, recent studies have not performed as well as they should in reproducing these two aggregations. Please see Tables 7 and 10 of the original paper.)
>
> > *Weakness:*  Regarding the description of Table 3 and 4, I believe it can be expressed in a more detailed manner than several highlights as bullet points in the current version. Specifically, different configurations are selected (pre-trained model and training set), and as a reader and VPR practitioner, I would be interested in analysis with respect to these choices.
>
> **A:** Thank you for your further explanation. We appreciate this suggestion, and we plan to update a preliminary version before the discussion is over. We will continue to improve this section before the official ICLR results are announced.
>
> > *Weakness:* I do not see this as a fair comparison with models trained on smaller datasets excluding the test sequences' training data.
>
> **A:**
>
> **We are very concerned about the fairness of research.**
>
> L84: *With the same training set, these improved aggregations achieve comparable or even superior results to recent approaches while requiring lower dimensions and fewer parameters.*
>
> L314-L319: *Here, we emphasize the **fairness** of our experimental design: (1) In
> Tab. 3 and 4, we employs GPMS, which differs from the training sets used by other methods. This divergence reflects the contribution of SLA, and previous methods have also used varying training sets. (2) In Tab. 5 - 11, all experiments within each table are conducted on the **same** training set. For example, G2M and NVL-FT2 were trained on GSV-Cities, while SALAD was trained on GPMS in Tab. 10.*
>
> > *W:* I cannot agree that the presentation quality, sparseness, or differences in input dimensions of other approaches make their work irrelevant to the novelty claims.
>
> **A:**
> 1. **The first highlight** is two methods from nearly a decade ago that are better than recent novel approaches. Then, we make some simple but effective changes (adding linear layers). **The second highlight** of our work is Supervised Label Alignment (SLA) for VPR, which is a core factor in achieving the top rank on the MSLS-challenge leardboard.
>
> 2. Novelty is unrelated to presentation quality or sparsity. **We intended to convey that** the sparsity in their descriptions makes it easy to judge the relevance of their methods to the innovations presented in our paper.
> - [1]: "The dimension reduction is performed using PCA with whitening followed by L2-normalization." (L208: "*employing PCA for dimension reduction.*")
> - [2]: Its experimental results contradict ours, further validating our contributions.
> - [3]: The cited module is a feature fusion module, while ours is a channel attention module, addressing distinct problems.

---

> ### Author Response · Authors · 2024-11-23
>
> > *Weakness:* While the simple modifications ..., it is likely that other methods have done something similar earlier.
>
> **A:** 1. We are very concerned about the literature reviews of research.
>
> 2. This statement seems to contradict the only strength about our paper that you summarized: *The paper extensively and thoroughly presents the historical overview of the VPR field (introduction and related works sections).*
>
> 3. To our knowledge, there is no prior work that specifically addresses this problem in the way we have approached it. (At least 5 researchers with more than 5 years of VPR research experience have participated in the discussion of this work, hoping that this can alleviate your concern.)
>
> 4. Behind these simple modifications you see are a lot of attempts and thoughts that you cannot see.
>
> 5. If such work exists and is as effective as our implementation, then they should be compared with recently published papers.
>
> 6. The simplicity of an idea is often confused with a lack of novelty when exactly the opposite is often true. [6]
>
> > *Weakness:* Concerning the quantization in SLA, my comment was referring to the possibility of conducting an ablation study on the underlying parameters such as M and $\alpha$
>  for determining the place ID and position, since testing is performed with the conventional protocols of GPS-coordinate-based average recall metrics.
>
> **A:** L259: We follow the quantization setup proposed by Berton et al. (2022a). For more details, please see Table 5 on page 8 of their paper [5].
>
> ```
> [1] Gong et al. "Ghost-dil-NetVLAD: A Lightweight Neural Network for Visual Place Recognition", 2024
> [2] Griange et al. "Design Space Exploration of Low-Bit Quantized Neural Networks for Visual Place Recognition", 2023
> [3] Wu et al. "Asymmetric Feature Fusion for Image Retrieval", 2024
> [4] Berton, Gabriele, et al. "Deep visual geo-localization benchmark." Proceedings of the IEEE/CVF Conference on CVPR. 2022.
> [5] Berton, Gabriele, et al. "Rethinking visual geo-localization for large-scale applications." Proceedings of the IEEE/CVF Conference on CVPR. 2022.
> [6] Michael Black. Novelty in Science.
> ```
>
> **Thank you again for the great efforts and valuable comments.**

---

> ### Author Response · Authors · 2024-11-26
>
> **Thank you for your feedback and for updating us on the score.**
>
> > *Concern:* writing and consistent story-line, where two different VPR methods with small architectural modifications are proposed.
>
> **A**: Firstly, I’d like to clarify a few points:
>
> 1. As the first author, I am satisfied with the current storyline.
>
> 2. Other reviewers have noted strengths (scores) in our presentation:
>
>     - "the organization and presentation of this paper are good." (YFv3)
>
>     - "good presentation", "simple and elegant." (snqV)
>
>     - "fair presentation" (ZnHW)
>
> Secondly, **we value your comments.** While I am unable to fully revise the storyline during the discussion phase, I will discuss this matter with my co-authors and ensure it is improved in the final version. Below is a proposed storyline for refinement:
>
> Classic feature aggregation methods, such as NetVLAD and GeM, operate in distinct feature dimension ranges (>10,000 and <2,000, respectively). Recently, approaches like SALAD have explored an intermediate dimensional range while maintaining the characteristic equation $f=a(x) \cdot b(x)$ from NetVLAD. This motivated us to explore:
>
> - How to compress NetVLAD into a similar dimensional range while retaining its high-dimensional performance.
> - How to introduce the equation characteristic to extend GeM's representational capacity.
>
> > *Concern:* reporting comparisons with SOTA in Tables 3-4 on the same training set to demonstrate the superiority of the method with the same input data in addition to existing results with the proposed GPMS dataset
>
> **A**: Thank you for your clarification, which has helped us fully understand the fairness issue you emphasized.
>
> This concern resonates with our earlier response (A.2.1 to reviewer ZnHW), and we appreciate the shared emphasis on fairness in comparisons.
>
> ```
> A.2.1: Most VPR research does not explicitly compare the use of different training datasets as we do in Table 3. While some datasets are tied to specific methods, this inherently poses challenges to the fairness of comparisons. Inconsistent labels have also been considered a limitation in previous study [2]. Therefore, label alignment can help solve these issues.
> ```
>
> Initially, Table 3 did include comparisons using the same training set. However, **after careful deliberation**, we decided to remove these. In the VPR field, similar comparisons in such tables typically focus on the best-performing settings. Considering variations in configurations (e.g., training datasets), direct comparisons may become unfair. However, comparing each method under its optimal settings ensures fairness. For instance:
>  - Comparing our method (trained on GSV-Cities) with SALAD and BoQ is fair.
>  - However, comparisons with EffoVPR or SelaVPR (trained on SF-XL or MSLS) could disadvantage our method.
>  - SF-XL-trained methods also use higher resolutions (e.g., 512x512), further complicating fairness.
>  - **Please see Table 3 of EffoVPR. (https://openreview.net/pdf?id=NSpe8QgsCB) Our Table 3 is more detailed than theirs. As for fairness, neither of them meets your requirements, but the EffoVPR has been recognized by multiple reviewers (6688).**
>
> This is why we included Tables 10 and 11, which separately compare our method with SALAD and BoQ.
>
> Anywhy, adding these experiments is simple. If, after considering our reasoning, you still find this comparison necessary, we are happy to include these experiments in Tab. 3.
>
> **(We have added these experiments in the updated version.)**
>
> > *Concern:* ablation studies on SLA parameters and reporting the results on "continuous coordinates for data alignment through triplet mining methods" experiments to strengthen the second contribution
>
> **A:**
>
> ~~We are currently working on these experiments, and they involve a significant engineering workload.~~
>
> ~~We will share results as soon as they are complete and include them in the final version of the paper.~~

---

> ### Author Response · Authors · 2024-11-27
>
> > *Concern:* ablation studies on SLA parameters and reporting the results on "continuous coordinates for data alignment through triplet mining methods" experiments to strengthen the second contribution
>
> **A**:
>
> 1. As mentioned earlier, SLA is a pre-processing algorithm. Its effectiveness can be directly validated by observing whether images of similar locations are correctly classified after parameter adjustments.
>
> 2. Similar to other works [1, 2] that align datasets, ablation studies on such aspects are rarely performed, as these experiments are often highly complex and challenging. Additionally, it is widely acknowledged that more data generally yields better results  (the scaling law).
>
> 3. We agree that additional experiments, as you suggested, could further reinforce our second contribution.
>
> We consistently use G (GSV-Cities) as the foundation dataset and conduct experiments with DINOv2-Base-G$^2$M using parameters as described in our paper, with a batch size of 128. We performed the following ablation studies:
>
> - **MSLS (M)**: Ablation on parameters M and N.
>
> - **Pittsburgh (P)**: Comparison of different alignment methods—the grid-based partitioning method and the triplet mining method.
>
> - **SF-XL (S)**: Scaling ablation to test the effect of increasing training data size.
>
> |    |   | Pitts-30k-test |      |      | MSLS-Val |      |      | SF-XL-Val |      |      |
> |----|---|----------------|------|------|----------|------|------|-----------|------|------|
> | M  | N | R@1            | R@5  | R@10 | R@1      | R@5  | R@10 | R@1       | R@5  | R@10 |
> | 5  | 5 | 91.6           | 96.3 | 97.5 | 91.0     | 95.8 | 96.5 | 89.9      | 94.8 | 96.4 |
> | 10 | 5 | 91.8           | 96.0 | 97.4 | 91.1     | 96.5 | 96.9 | **91.6**      | 96.2 | 97.3 |
> | 15 | 5 | 91.7           | 95.8 | 97.1 | **92.3**     | 96.6 | 97.2 | 91.4      | 95.8 | 96.9 |
> | 10 | 4 | 91.8           | 95.9 | 97.1 | 91.8     | 96.0 | 96.9 | 91.0      | 95.8 | 97.0 |
> | 10 | 5 | 91.8           | 96.0 | 97.4 | 91.1     | 96.5 | 96.9 | **91.6**      | 96.2 | 97.3 |
> | 10 | 6 | **92.0**           | 96.0 | 97.5 | 91.4     | 96.8 | 97.0 | 90.9      | 95.8 | 97.0 |
>
> |        | Pitts-30k-test |      |      | MSLS-Val |      |      | SF-XL-Val |      |      |
> |:------:|:--------------:|:----:|:----:|:--------:|:----:|:----:|:---------:|:----:|:----:|
> |        | R@1            | R@5  | R@10 | R@1      | R@5  | R@10 | R@1       | R@5  | R@10 |
> | SLA-GP | 91.2           | 95.6 | 97.0 | 90.7     | 94.9 | 95.5 | 89.8      | 95.0 | 96.4 |
> | SLA-TM | **91.4**       | 95.9 | 97.0 | **91.4**     | 96.0 | 96.6 | **90.3**      | 95.2 | 96.6 |
>
>
> |            | Pitts-30k-test |      |      | MSLS-Val |      |      | SF-XL-Val |      |      |
> |:--------:|:--------------:|:----:|:----:|:--------:|:----:|:----:|:---------:|:----:|:----:|
> | # Number | R@1            | R@5  | R@10 | R@1      | R@5  | R@10 | R@1       | R@5  | R@10 |
> | 60k      | **92.0**           | 96.0 | 97.0 | 90.1     | 95.8 | 96.9 | 91.5      | 96.1 | 97.2 |
> | 120k     | 91.6           | 95.8 | 97.0 | 90.8    | 95.4 | 96.1 | 91.2      | 96.0 | 97.2 |
> | 180k     | 91.7           | 96.0 | 97.1 |  **91.0**     | 95.5 | 96.4 | **93.1**      | 96.8 | 97.9 |
>
> Tab. 1: We observe that $M$ directly impacts the aligned data size. Using the full MSLS training set for alignment would diminish differences seen in these results.
>
> Tab. 2: While triplet mining (TM) achieves slightly better results than grid partitioning (GP), its code logic involves multiple nested loops, making it computationally infeasible for larger datasets.
>
> Tab. 3: These results align with scaling laws, showing that performance improves as training data increases.
>
>
> To expedite the ablation experiments, we only used a subset of the MSLS training set. This approach has been used in prior works, such as Table 3 of [3], for ablation studies.
>
> ```python
> city_names = ['amman', 'amsterdam', 'austin', 'bangkok', 'berlin', ] # 'boston', 'budapest', 'goa', ]
>             #  'helsinki', 'london', 'manila', 'melbourne', 'moscow', 'nairobi', 'ottawa', 'paris',
>             #  'phoenix', 'saopaulo', 'tokyo', 'toronto', 'trondheim', 'zurich']
>
> ```
>
> ```
> [1] Wang, Shuzhe, et al. "Dust3r: Geometric 3d vision made easy." Proceedings of the IEEE/CVF Conference on Computer Vision and Pattern Recognition. 2024.
> [2] Yang, Lihe, et al. "Depth anything: Unleashing the power of large-scale unlabeled data." Proceedings of the IEEE/CVF Conference on Computer Vision and Pattern Recognition. 2024.
> [3] Ali-bey, Amar, Brahim Chaib-draa, and Philippe Giguere. "Gsv-cities: Toward appropriate supervised visual place recognition." Neurocomputing 513 (2022): 194-203.
> ```
>
> **Thank you again for the great efforts and valuable comments.**
>
> **We hope that this provides sufficient reasons to further raise the score or your confidence.**

---

> ### Author Response · Authors · 2024-12-02
>
> **Dear Reviewer fhKJ,**
>
> We have carefully and in detail addressed your remaining concerns.
>
> We hope you might find the response satisfactory.
>
> May we kindly ask if you would consider raising your score by **1 point**?
>
> **Best,**
>
> *The authors*

---

### Official Review · Reviewer_snqV · 2024-11-01

**Soundness:** 3
**Presentation:** 3
**Contribution:** 3
**Rating:** 8
**Confidence:** 3

**Summary:**

This paper investigates how modern foundation model DINOv2 can be utilized for the task of visual place recognition. The paper has two main contributions. First, by revisiting the already existing feature aggregating methods with some minor tweaks, the paper observes that the resulting methods achieve very competitive results while in many cases being simpler than previous methods. Second, the paper proposes a method to chain multiple place recognition datasets into one to benefit from large-scale training. At last, the paper thoroughly evaluates and ablate their main components of the proposed method on multiple datasets and in comparison to other methods.

**Strengths:**

Overall, the contributions of this paper are interesting and worth publication, in particular, revisiting time-tested methods, improving them in a clever way, and showing that with a few small changes, they can still achieve SOTA performance.

To summarize:
- The proposed method is simple and elegant, yet achieves strong performance on multiple benchmarks.

- The proposed method improves widely used GeM and NetVLAD pooling layers for use with DINOv2 foundation model.

- The paper proposes a scheme to chain the training datasets together and it seems to be outperforming the recently published method (SALAD) to which authors compare.

- The different components of the proposed method are thoroughly ablated.

- The resulting descriptors are relatively compact (768D for SP-G2M model and 8192D for SP-NVL-FT2 model) making the method applicable for large-scale applications.

**Weaknesses:**

The biggest issue is in the way the paper defines the supervised label alignment technique as well as some training details. While this contribution seems to be an important part of the paper (as shown in Table 6), it is not really clear how it works. The only bits of information can be found in Table 1, but it is not clear how to interpret it. Is the SimplePlace model trained on each dataset separately with different label schemes and loss functions? Or is it chaining all the datasets into one by finding a labeling scheme that can be derived from all of them? Adding more details and explaining this in the rebuttal would be beneficial.

**Questions:**

1. While I understand that this method is based on DINOv2, is there a way on how to test the generalizability of G2M and FT2 approaches on other foundation models?

2. Can you please clarify step-by-step how supervised label aligning works? What loss function is used in the training?

3. Would it be also possible to report results (on at least one dataset) with DINOV2 and the unmodified NetVLAD and GeM pooling methods?

---

> ### Author Response · Authors · 2024-11-17
> **Response**
>
> **Thank you for your thoughtful comments and suggestions. We hope the following clarifications address your concerns.**
>
> > *Question:* While I understand that this method is based on DINOv2, is there a way on how to test the generalizability of G2M and FT2 approaches on other foundation models?
>
> **A:** Thank you for the insightful suggestion. We have fine-tuned the CLIP visual encoder to test generalizability, given that CLIP is a well-known foundation model. The results were consistent with those obtained using DINOv2, demonstrating that the contributions of our paper generalize well to other foundation models.
>
> | Class      | Model | Aggregation    | Dimension | Training Set | MSLS-Val R@1 | MSLS-Val R@5 | Pitts-30k R@1 | Pitts-30k R@5 | SF-XL Val R@1 | SF-XL Val R@5 |
> |------------|-------|----------------|-----------|--------------|--------------|--------------|---------------|---------------|---------------|---------------|
> | Baseline-3 | CLIP  | GeM            | 768       | G            | 79.2         | 88.8         | 86.8          | 94.7          | 81.1          | 89.4          |
> |            | CLIP  | GeM            | 768       | GPMS         | 85.3         | 93.1         | 88            | 94.8          | 83            | 91.1          |
> |            | CLIP  | G2M (Ours)     | 768       | GPMS         | 86.1         | 93.5         | 89.1          | 95            | 85.4          | 92.7          |
> | Baseline-4 | CLIP  | NetVLAD        | 49152     | G            | 82.8         | 91.1         | 89.3          | 95.4          | 83.7          | 90.9          |
> |            | CLIP  | NetVLAD        | 49152     | GPMS         | 88.1         | 93.7         | 90.2          | 95.7          | 86.7          | 93.7          |
> |            | CLIP  | NVL            | 8192      | GPMS         | 86.4         | 93.4         | 89.6          | 95.4          | 84.8          | 92.7          |
> |            | CLIP  | NVL-FT2 (Ours) | 8192      | GPMS         | 87           | 93.4         | 89.9          | 95.5          | 86.1          | 93            |
>
> > *Question:* Can you please clarify step-by-step how supervised label aligning works? What loss function is used in the training?
>
> **A:** Thank you for your question. Here's a detailed breakdown of how supervised label alignment is implemented:
>
> 1. The datasets G, P, M, and S are mixed together during training and share the label format of GSV-Cities.
> 2. The loss function used is Multi-Similarity Loss (MS-Loss), which is mentioned in Line 161 of the paper.
> 3. The basic steps for label alignment are outlined in Section 3.3 of the paper, though some of the details are more implementation-specific (for example, refer to the supplementary materials and the utils/format_xx.py code). A core explanation (L263) of Label Aligning is as follows:
>     - UTM coordinates (easting and northing) are rounded and grouped into class and group IDs. This allows the creation of meaningful labels for training.
>     - Specifically, the code snippet below illustrates the process of generating class_id and group_id from UTM coordinates:
>
>         ```python
>             @staticmethod
>             def get__class_id__group_id(utm_east, utm_north, M = 10, N = 5):
>                 """Return place_id (class_id + grounp_id) for a given point.
>                 """
>                 rounded_utm_east = int(utm_east // M * M)  # Rounded to nearest lower multiple of M
>                 rounded_utm_north = int(utm_north // M * M)
>
>                 class_id = (rounded_utm_east, rounded_utm_north)
>                 # group_id goes from (0, 0) to (N, N)
>                 group_id = (rounded_utm_east % (M * N) // M,
>                             rounded_utm_north % (M * N) // M)
>                 return class_id, group_id
>         ```
>
>     - This method creates aligned labels by mapping spatial coordinates to structured IDs, making the training efficient and consistent across large datasets.

---

> ### Author Response · Authors · 2024-11-17
> **Response 2**
>
> > *Question:* Would it be also possible to report results (on at least one dataset) with DINOV2 and the unmodified NetVLAD and GeM pooling methods?
>
> **A:** We appreciate your feedback, and we will consolidate this information and add it to the supplementary materials for clarity.
>
> | Class      | Model  | Aggregation    | Dimension | Training Set | MSLS-Val R@1 | MSLS-Val R@5 | Pitts-30k R@1 | Pitts-30k R@5 |
> |------------|--------|----------------|-----------|--------------|--------------|--------------|---------------|---------------|
> | Baseline-5 | DINOv2 | GeM            | 768       | G            | 90.3         | 95.4         | 91.9          | 96.6          |
> |            | DINOv2 | GeM            | 768      | GPMS         | 91.5         | 96.4         | 92.4          | 96.8          |
> |            | DINOv2 | G2M (Ours)     | 768     | GPMS         | 92.4         | 96.8         | 92.0          | 96.6          |
> | Baseline-6 | DINOv2 | NetVLAD        | 49152     | G            | 93.0         | 96.7         | 93.0          | 96.7          |
> |            | DINOv2 | NetVLAD        | 49152     | GPMS         | 94.6         | 97.6         | 93.5          | 97.4          |
> |            | DINOv2 | NVL            | 8192      | GPMS         | 93.8         | 97.3         | 93.0          | 97.1          |
> |            | DINOv2 | NVL-FT2 (Ours) | 8192      | GPMS         | 94.6         | 97.8         | 93.1          | 97.4          |
>
>  We sincerely appreciate your constructive suggestions and believe that the additional experiments, analysis, and explanations significantly improve the quality of our submission.
>
> **We hope that this provides sufficient reasons to raise the score.**

---

> ### Author Response · Authors · 2024-11-22
>
> Dear Review snqV,
>
> Thank you again for the great efforts and valuable comments. We have carefully addressed the main concerns in detail. We hope you might find the response satisfactory. As the discussion phase is about to close, we are very much looking forward to hearing from you about any further feedback. We will very happy to clarify any further concers (if any).
>
> Best, Authors

---

> ### Comment · Reviewer_snqV · 2024-12-01
> **Update of the review and score**
>
> Dear Authors,
>
> Thank you for the provided responses. The response is addressing my main concerns. I raise the score to 8.
>
> Reviewer

---

> > ### Author Response · Authors · 2024-12-01
> >
> > Thank you for your positive feedback and for updating us on the score.
> >
> > We genuinely appreciate your insightful and constructive comments.

---

### Official Review · Reviewer_YFv3 · 2024-11-02

**Soundness:** 3
**Presentation:** 3
**Contribution:** 2
**Rating:** 3
**Confidence:** 4

**Summary:**

This paper presents two kinds of foundation model based compact descriptors for visual place recognition. In this paper, based on two classical GEM and NetVlad descriptors, the authors proposed the corresponding variants. In addition, a label alignment strategy is proposed to be tailored for the foundation vision model pre-training. Extensive experiments are conducted on 12 test datasets to demonstrate the effectiveness of the proposed framework.

**Strengths:**

1. The paper is readable, where the organization and presentation of this paper are good.
2. Extensive experiments are conducted to verify the proposed framework.

**Weaknesses:**

1. The main concern on this paper is that the contribution of this paper is incremental. With the pre-trained DINO-v2 foundation model, the authors proposed two variants of GEM and Netvlad, respectively. By introducing an additional branch to project GEM into the low-dimensional space with MLP, the G^2M descriptor is proposed. By fine-tuning the backbone network and linear layer of Netvlad, the NVL-FT^2 descriptor is developed. In fact, the training strategies are usually employed in the design of the network structure. Moreover, this paper lacks deep analysis of the core modules in the proposed framework, i.e.,  the reasons why the simple operations on the classical visual place recognition descriptors can lead to significant improvement.
2. What is the relationship between the proposed G^2M descriptor and NVL-FT^2 descriptor? It seems that the proposed method is not self-contained. The same insights are conveyed to readers from the proposed descriptors? Therefore, I am confused that why the authors proposed two independent variants for the visual place recognition task.
3. The authors claim that the proposed G^2M descriptor is suitable for the real-time applications while the NVL-FT^2 descriptor is suitable for high-performance cases. However, in the experimental section, I did not see the comparison of the inference time of the G^2M descriptor to verify the real-time performance. In addition, the idea case is that a real-time and high-performance visual place recognition descriptor is desired. How to deal with the case with the proposed variants?
4. In this paper, there are too many abbreviations, which might be confused for readers. For example, in Line 417, what is SLA?

**Questions:**

Please see the weakness section.

---

> ### Author Response · Authors · 2024-11-17
> **Response to W1, 2, 3, 4**
>
> **Thank you for your thoughtful comments and suggestions. We hope the following clarifications address your concerns.**
>
> > *Weakness:* The main concern on this paper is that the contribution of this paper is incremental.
>
> **A:** We acknowledge that these two contributions are incremental, but ...
>
> 1. **The first highlight** is two methods from nearly a decade ago can be better than recent **novel** approaches. (papers accepted by CVPR'24, ECCV'24, and ICLR'24, and 2 papers under reviewed at ICLR'25) Then, we make some simple but effective changes (adding linear layers).
>
> 2. **The second highlight** of our work is Supervised Label Alignment (SLA) for VPR, which is a core factor in achieving the top rank on the MSLS-challenge leardboard.
>
> 3. Revisiting time-tested methods, improving them in a clever way, and showing that with a few small changes, they can still achieve SOTA performance. (Reviewer snqV)
>
> > *Weakness:* The reasons why the simple operations on the classical visual place recognition descriptors can lead to significant improvement.
>
>   1. The GCA of G2M learns the principal components of the feature maps along the channel dimension to calibrate the GeM pooling vector accordingly.
>   2. The NVL output features are used directly for loss and gradient calculations, meaning the optimization process occurs in an explicitly low-dimensional space (but with implicit high-dimensional space). The second-order fine-tuning for NVL achieves the best fit in both high-dimensional and low-dimensional spaces.
>
> L229: Given that both methods operate with the same feature dimension, training set, and neural network
> architecture, such a performance gap is unexpected. Intuitively, the linear projection should outperform PCA. The key difference, however, lies in their training methodologies.
>
> > *weakness:* I am confused that why the authors proposed two independent variants for the visual place recognition task.
>
> > *Weakness:* The idea case is that a real-time and high-performance visual place recognition descriptor is desired. How to deal with the case with the proposed variants?
>
> **A:** The issues raised in W2 and W3 are somewhat overlapping, so we address them together here, given the complexity and practical relevance of the matter.
>
> Firstly, let’s revisit the definition of VPR: retrieving images from a large-scale geographic image database that are visually similar to a query image. The scale of the database significantly influences the difficulty of retrieval, as well as the storage and time requirements. For instance, a NetVLAD generates feature vectors of dimension 32k, which, for a database of 10 million images, would require approximately 32k × 10M × 4B = 1220GB of memory [1]. Retrieval times on the SF-XL and Pitts-30k test set are shown in the below table.
>
> |           | Dimension | Dataset name | # Database | Avg Test time |
> |-----------|-----------|--------------|------------|---------------|
> | G2M       | 768       | Pitts-30k    | 10000      | 0.43ms        |
> | NV-Linear | 8192      | Pitts-30k    | 10000      | 4.2ms         |
> | NetVLAD   | 49152     | Pitts-30k    | 10000      | 25.9ms        |
> | G2M       | 768       | SF-XL        | 2.8M       | 123.4ms       |
> | NV-Linear | 8192      | SF-XL        | 2.8M       | X             |
> | NetVLAD   | 49152     | SF-XL        | 2.8M       | X             |
>
> Although our experimental equipment has 120GB of memory, NVL and NetVLAD cannot utilize FAISS [3] on SF-XL test set. They require a more memory-efficient approach for computation. (Please refer to utils/test.py)
>
> While there is a significant difference in retrieval times between low-dimensional and high-dimensional features for large-scale databases, and [1] considers large databases as a real-world challenge, we raise the question: **do real-world applications always require retrieval from such vast databases?**
>
> Consider two real-world VPR scenarios [2]: robotics and augmented reality (e.g., on smartphones). In practice, both use cases can leverage network or GPS to obtain a prior location estimate, significantly narrowing the database search scope.
>
> We understand both perspectives and thus have retained the original statement: G2M excels in large-scale applications requiring rapid response and low latency, while NVL-FT2 is optimized for scenarios demanding high precision across a broad range of conditions.
>
> **Update on 23 Nov:**
>
> *L74: However, through some tentative experiments, we found that classic methods from ten years ago are still competitive. This prompted us to improve these classic methods instead of following recent aggregations introduced in the past year.*
>
> This statement also implies the explanation of why we improved both aggregation layers simultaneously.

---

> ### Author Response · Authors · 2024-11-17
> **Reference**
>
> > *weakness:* I did not see the comparison of the inference time of the G^2M descriptor to verify the real-time performance.
>
>  **A:** **In fact, we report the comparison of inference time in Table 11.** We present the following comparisons in the table, covering the number of parameters in the aggregation layer, single-feature aggregation inference speed, and feature dimension. Our two improved methods demonstrate clear advantages in parameter count and feature dimension. In terms of speed, G2M leads with 0.41 ms, while NVL-FT2 shows a slight disadvantage due to the loop implementation in residual aggregations.
>
> Considering Recall@K alone, BoQ slightly outperforms NVL-FT2. However, factoring in BoQ's complex training techniques, extended training time, larger parameters, and higher feature dimensions, NVL-FT2 offers a more efficient solution.
>
> | Method          | Param. (M) | Infer. time (ms) | Feat. dim | Pitts-30k-test R@1 | Pitts-30k-test R@5 | MSLS-val R@1 | MSLS-val R@5 |
> |-----------------|------------|------------------|-----------|--------------------|--------------------|--------------|--------------|
> | BoQ  (CVPR 24)  | 8.63       | 2.53             | 12288     | 93.7               | 97.1               | 93.8         | 96.8         |
> | SALAD (CVPR 24) | 1.41       | 1.45             | 8448      | 92.4               | 96.3               | 92.4         | 96.2         |
> | NVL-FT2 (Ours)  | 0.197      | 4.49             | 8192      | 93.4               | 97.0               | 93.1         | 96.6         |
> | G2M (Ours)      | 0.69       | 0.41             | 768       | 92.6               | 96.8               | 90.4         | 95.9         |
>
> **The methods in the above table use the same training set and pre-trained model.**
>
>
> > *weakness:* In this paper, there are too many abbreviations, which might be confused for readers. For example, in Line 417, what is SLA?
>
>  **A:** SLA is the abbreviation of Supervised Label Alignment. Thank you for your suggestion, I will change the SLA in the section title to the full name.
>
> ```
> [1] Berton, Gabriele, et al. "Rethinking visual geo-localization for large-scale applications." Proceedings of the IEEE/CVF Conference on Computer Vision and Pattern Recognition. 2022.
> [2] Sattler, Torsten, et al. "Benchmarking 6dof outdoor visual localization in changing conditions." Proceedings of the IEEE/CVF Conference on Computer Vision and Pattern Recognition. 2018.
> [3] Douze, Matthijs, et al. "The faiss library." arXiv preprint arXiv:2401.08281 (2024).
> ```
>
>  We sincerely appreciate your constructive suggestions and believe that that the additional experiments, analysis, and explanations significantly improve the quality of our submission.
>
> **We hope that this provides sufficient reasons to raise the score or your confidence.**

---

> ### Author Response · Authors · 2024-11-22
>
> **Update on 23 Nov:**
>
> We use FAISS (Facebook AI Similarity Search) for efficient similarity search and clustering of dense vectors. Its computational complexity depends on several factors, including the feature dimension $d$, the database size $n$, and the indexing method used. Consistent with prior work [1], we utilize FAISS's Brute-Force Search, which computes pairwise distances between the query vector and all database vectors. For a single query, the complexity of computing distances between $n$ database vectors, each of dimension $d$, is: $O(n \cdot d)$.
>
> G$^2$M is highly efficient for large-scale ($n$ is large) applications requiring rapid response and low latency.
>
> NVL-FT$^2$, with a higher dimensionality $d$, is better suited for tasks requiring high precision across diverse conditions.
>
> This differentiation demonstrates the adaptability of two aggragations to meet diverse requirements.
>
> We hope you might find the response satisfactory, and we apologize if this reminder email bothers you.
>
> **Update on 25 Nov**
>
> > *Weakness:* The reasons why the simple operations on the classical visual place recognition descriptors can lead to significant improvement.
>
> **A:** In addition to the previous explanations, these designs also contain our experience distilled from previous research. Both NetVLAD and SALAD involve operations that can be interpreted as $f  = a (x) \cdot b (x)$, and we realize that the low-dimensional feature method (i.e. GeM) lacks such operations.
>
> **NetVLAD:**
>
> 1. **Soft assignment $a$:**
>    The cluster assignment weights are denoted as $a_{n, c, i} $, where:
>    - $n$: Batch index.
>    - $c$: Cluster index.
>    - $i $: Spatial location index in the feature map (e.g., over $H \times W $).
>
>    The soft assignment is computed as: $a_{n,c,i}$ = softmax(conv($x$)), where conv($x$) represents the raw cluster assignment score at spatial location.
>
> 2. **Residuals $b$:**
>    The residuals $b_{n, c, i}$ are defined as: $b_{n,c,i}$ = $x_{n,:,i} - $centroid$_c$
>
>    where:
>    - $x_{n,:,i}$: The feature vector at location $i$.
>    - centroid$_c$: The centroid of the cluster $c$.
>    This computes the difference between the feature and its assigned cluster center.
>
> 3. **Aggregation $ f $:**
>    The aggregated feature for cluster $ c $ in sample $ n $ is: $f_{n,c}$ = $\sum_{i}a_{n,c,i} \cdot b_{n,c,i}$
>
>    Here:
>    - $ a_{n,c,i} $: Cluster assignment weight at location $ i $.
>    - $ b_{n,c,i}$: Residual vector at the same location.
>
> **SALAD:**
>
> 1. **Cluster assignment $ a $:**
>    After applying the Sinkhorn algorithm, the soft assignment weights $ a_{n, m, k} $ are derived, where:
>    - $ n $: Batch index.
>    - $ m $: Cluster index.
>    - $ k $: Flattened spatial location index (over $ H \times W $).
>
>    These assignments are computed as: $a_{n,m,k} = p_{n,m,k} $= exp$($Sinkhorn$(S)_{n,m,k})$
>
>    Here:
>    - $ S_{n,m,k} $: The score matrix value for cluster $ m $ and location $ k $.
>    - The $ a_{n,m,k} $ values represent the weight assigned to location $ k $ for cluster $ m $.
>
> 2. **Cluster features $ b $:**
>    The features $ b_{n, :, k} $ are extracted as: $b_{n,:,k} = f_{n,:,k}$ = MLP($x)_{n,:,k}$
>
>    where:
>    - $ b_{n,:,k} $: The feature vector at location $ k $.
>
> 3. **Aggregation $ f $:**
>    For each cluster $m$, the aggregated feature is: $f_{n,m} = \sum_k a_{n,m,k} \cdot b_{n,:,k}$
>
>    where:
>    - $ a_{n,m,k} $: Cluster assignment weight.
>    - $ b_{n,:,k} $: Feature vector at location $k$.
>
> 4. **Global representation:** (option)
>    The final global descriptor concatenates the normalized scene token $t_n$ with the aggregated cluster features:
>
> $f_n = $normalize$(t_n) \oplus $normalize$(\sum_k a_{n,m,k} \cdot b_{n,:,k})$
>
> **G$^2$M (Ours):**
>
> 1. **Generalized Channel Attention $ a $:**
>
> The attention mechanism is applied to the features, producing a feature vector:
> $x_{atte}$ = GCA$(x)$.
>
> 2. **GeM Pooling and Linear $ b $**:
>
> The features is passed through GeM (Generalized Mean Pooling) and a linear layer:
>
>   $x_{feat}$ = Linear(GeM$(x))$
>
> 3. **Aggregation $ f $:**
>
> The features are multiplied by the attention weights:
>   $f = x_{feat} \cdot x_{atte}$.
>
>
> **Thus, in all these methods**, the aggregation can be interpreted as involving an operation $f = a(x) \cdot b(x) $, where $a $ and $ b $ are different parts of the features (residuals, attention weights, or probability matrices) that are multiplied together to produce the final aggregated representation.

---

> ### Author Response · Authors · 2024-11-23
>
> **Dear Review YFv3**,
>
> Thank you again for the great efforts and valuable comments. We have carefully addressed the main concerns in detail. We hope you might find the response satisfactory. As the discussion phase is about to close, we are very much looking forward to hearing from you about any further feedback. We will very happy to clarify any further concers (if any).
>
> **Best,**
> *Authors*

---

### Official Review · Reviewer_ZnHW · 2024-11-02

**Soundness:** 3
**Presentation:** 2
**Contribution:** 3
**Rating:** 6
**Confidence:** 4

**Summary:**

The paper presents SuperPlace, a VPR model that improves classic feature aggregation techniques like GeM and NetVLAD. Two main improvements are introduced: (1) G2M, a compact two-layer GeM module designed to optimize feature maps along the channel direction, and (2) a secondary fine-tuning method (FT2) for NetVLAD-Linear, which refines high-dimensional representations before compressing them. Additionally, a supervised label alignment method is proposed to facilitate multi-dataset training, inspired by Visual Foundation Models. Extensive benchmarking indicates that SuperPlace achieves top results on several VPR datasets while using fewer feature dimensions.

**Strengths:**

1. The paper demonstrates that classical aggregation techniques, when carefully refined, can match or outperform modern complex methods. This insight is valuable for the community to reconsider the recent trend of utilizing VLMs for VPR tasks.
2. The authors provide a thorough evaluation across multiple benchmark datasets, comparing against recent state-of-the-art VPR methods.

**Weaknesses:**

1. The contribution of the two proposed methods are largely incremental. The channel attention module is widely used to aggregate information in different channels, as is in the SE paper that the authors already cited. I don't see any difference of the proposed module to previous practices, and I don't think that the "motivations, usage, and design details are novel." Also, two-stage training is a very common practice in fine tuning pretrained models.
2. It is not clear on necessity of label alignment. If using triplet loss for supervision, the only information needed is the positive and negative sets for each query image, which can be easily determined by applyting thresholds on the location and direction.
3. Although the authors demonstrated the superior performance with the DINOv2 representation, there is no experiments showing what is the improvement for other backbones like VGG, ResNet, and ViT. This significantly weakens the generalizability of the proposed method on different backbones and representations.
4. While I appreciate the insights provided in Figure 2, it is not discussed in depth. If the location information is dominant in the first component, why not simply empoly it?
5. The illustration in Figure 1 is confusing. For example, different colors are used for the same component (patch tokens, feature map). Same color is used for different component (GeM, VFM, SoftMax, etc.). It is also not clear what's the difference between the two discriptors in NVL-FT^2.

**Questions:**

I appreciate the authors' insight on why NV-Linear doesn't give exptected performance, but this is not very convincing. Why does the training of NV-Linear happens in a "a lower-dimensional space"?

If the above question is answerd and concerns addressed, I'm happy to raise my rating.

---

> ### Author Response · Authors · 2024-11-16
> **Response to W1, 2**
>
> **Thank you for your thoughtful comments and suggestions.**
>
> **We hope the following clarifications address your concerns.**
>
> > *Weakness*: The contribution of the two proposed methods are largely incremental. The channel attention module is widely used to aggregate information in different channels, as is in the SE paper that the authors already cited. I don't see any difference of the proposed module to previous practices, and I don't think that the "motivations, usage, and design details are novel." Also, two-stage training is a very common practice in fine tuning pretrained models.
>
> **A:** We acknowledge that these two contributions are incremental. **The first highlight** is two methods from nearly a decade ago that are better than recent novel approaches. (To put it bluntly, recent studies have not performed as well as they should in reproducing these two works. Please see Tables 7 and 10 of the original paper.) Then, we make some simple but effective changes (adding linear layers).
>
> However, we respectfully disagree with the last two points in your assessment and would like to provide the following clarifications:
>
> 1. The differences between SE and GCA are as follows:
>     - The motivation for GCA stems from the visualization experiments of DINOv2.
>     - GCA's pooling and activation functions differ from SE's. While the differences may appear subtle, they are distinct.
>     - The SE module is designed for each network layer except for the final aggregation layer, making its usage fundamentally different.
>     - As shown in Table 7, GCA outperforms the SE module in performance. (The reason is in line 184.)
>     - We will revise the statement to: "The motivation, usage, and design details differ from those of the SE module."
>
> 2. Regarding the two-stage training, we provide the following points:
>     - Considering the pre-trained models mentioned, the complete process involves three-stage training.
>     - Our second-order fine-tuning focuses on the aggregation layer. We do not claim this to be a major innovation because it could be considered a special case of greedy layer-wise training [1].
>
>
> > *Weakness*: It is not clear on necessity of label alignment. If using triplet loss for supervision, the only information needed is the positive and negative sets for each query image, which can be easily determined by applyting thresholds on the location and direction.
>
>
> **A:** **The second highlight** of our work is Supervised Label Alignment (SLA) for VPR, which is a core factor in achieving the top rank on the MSLS-challenge leardboard. Our response regarding the necessity of SLA and another implementation is as follows:
> 1. Most VPR research does not explicitly compare the use of different training datasets as we do in Table 3. While some datasets are tied to specific methods, this inherently poses challenges to the fairness of comparisons. Inconsistent labels have also been considered a limitation in previous study [2]. Therefore, label alignment can help solve these issues.
> 2. SLA draws inspiration from scaling laws observed in the LLM field and aligns with recent trends in the research community [3][4]. Although Ilya Sutskever recently stated that scaling laws for LLMs may have reached their limits, we believe the VPR field is still far from exhausting this potential.
> 3. We appreciate the insight regarding the triplet mining method. Indeed, our earliest implementation followed the approach you described. (While the GitHub repository is currently private due to the double-blind policy, the earliest implementation will be accessible in the commit history once the repository is made public.) However, this method has limitations when applied to cropped images derived from panoramic sources. A notable issue arises when anchor and positive samples share the same GPS coordinates but do not share a common view. We also explored two potential solutions to this problem:
>     - Dynamic Ranking Method [5]: Effective but significantly increases training time.
>     - Local Feature Matching for Verification: Involves complex loop logic but addresses the view inconsistency issue.
>
> Ultimately, SLA emerged as a more concise and flexible solution. Importantly, the results from both implementations are consistent, but SLA's streamlined logic makes it easier to extend and adapt.

---

> ### Author Response · Authors · 2024-11-16
> **Response to W3, 4, 5**
>
> > *Weakness*: Although the authors demonstrated the superior performance with the DINOv2 representation, there is no experiments showing what is the improvement for other backbones like VGG, ResNet, and ViT. This significantly weakens the generalizability of the proposed method on different backbones and representations.
>
>
> **A:** VGG and ResNet are both convolutional architectures, and the theoretical structures of DINOv2 and ViT are identical, differing primarily in their improved model parameters. Therefore, we believe it is sufficient to conduct generalization experiments using ResNet-50. Additionally, another reviewer emphasized the importance of generalization experiments with other foundation models. To address this, we also trained using CLIP's visual encoder. Please refer to our response to that reviewer for further details. As shown in the table below, our method achieves the same conclusions on ResNet-50 as reported in the original paper.
>
> Specifically, the experimental setups for the baselines closely follow those in [6], but our implementation achieves better results. We believe this enhances the robustness and credibility of our supplementary experiments.
>
> | Class          | Model    | Aggregation    | Dimension | Training Set | MSLS-Val R@1 | MSLS-Val R@5 | Pitts-30k R@1 | Pitts-30k R@5 | SF-XL Val R@1 | SF-XL Val R@5 |
> |----------------|----------|----------------|-----------|--------------|--------------|--------------|---------------|---------------|---------------|---------------|
> | Baseline-1 [6] | ResNet50 | GeM            | 2048     | G            | 76.5         | 85.7         | -             | -             | -             | -             |
> | Baseline-1     | ResNet50 | GeM            | 1024      | G            | 84.5         | 90.5         | 89.8          | 94.8          | 87.7          | 93.3          |
> |                | ResNet50 | GeM            | 1024      | GPMS         | 87.6         | 92.8         | 90.4          | 95.3          | 90.1          | 94.8          |
> |                | ResNet50 | G2M (Ours)     | 1024      | GPMS         | 88.8         | 93.2         | 90.4          | 94.8          | 88.3          | 93.8          |
> | Baseline-2     | ResNet50 | NetVLAD        | 65536     | G            | 78.9         | 87.7         | 89.9          | 95            | 84.9          | 92            |
> |                | ResNet50 | NetVLAD        | 65536     | GPMS         | 89.2         | 93.8         | 90.4          | 95.3          | 88.9          | 94.1          |
> |                | ResNet50 | NVL            | 8192      | GPMS         | 87.6         | 93.8         | 89.9          | 94.9          | 88.2          | 93.7          |
> |                | ResNet50 | NVL-FT2 (Ours) | 8192      | GPMS         | 88           | 93.3         | 90.6          | 95.2          | 88.2          | 93.8          |
>
>
>
> > *Weakness*: While I appreciate the insights provided in Figure 2, it is not discussed in depth. If the location information is dominant in the first component, why not simply empoly it?
>
> **A:** We appreciate this insight. Our earliest implementation was as you described, but it did not lead to direct improvements in performance. We believe this is because PCA computes principal components highly correlated with the dataset. Unlike GCA, PCA cannot be easily integrated into a network for training on large-scale datasets.
>
> > *Weakness*:The illustration in Figure 1 is confusing. For example, different colors are used for the same component (patch tokens, feature map). Same color is used for different component (GeM, VFM, SoftMax, etc.). It is also not clear what's the difference between the two discriptors in NVL-FT^2.
>
> **A:** We share some of your concerns about the potential for confusion when using too many or too few colors. However, we respectfully disagree with parts of your example:
>
> 1. Patch tokens, feature maps, and descriptors are shown in the same color because they all represent in/output data across stages. This is a deliberate choice to emphasize their role as processed data entities.
> 2. Components like GeM, VFM, and SoftMax in NetVLAD are uniformly colored blue, as they originate from related prior work and share a functional context.
> 3. In summary, our approach to color usage applies the same color to components sharing similar attributes or roles.
> 4. We will consider refining the figure further to enhance clarity, especially regarding the difference between the two descriptors in NVL-FT2.

---

> ### Author Response · Authors · 2024-11-16
> **Response to Q1**
>
> > *Question*: Why does the training of NV-Linear happens in a "a lower-dimensional space"?
>
> **A:** The output dimension of NVL (8192) is significantly lower than that of NV (49152), indicating a lower-dimensional representation. The NVL output features are used directly for loss and gradient calculations, meaning the optimization process occurs in an explicitly low-dimensional space (but with implicit high-dimensional space). In contrast, NV operates directly in a high-dimensional space, where the optimization leverages a larger feature space for representation.
>
> ```
> [1] Bengio, Yoshua, et al. "Greedy layer-wise training of deep networks." Advances in neural information processing systems 19 (2006).
> [2] Berton, Gabriele, et al. "Rethinking visual geo-localization for large-scale applications." Proceedings of the IEEE/CVF Conference on Computer Vision and Pattern Recognition. 2022.
> [3] Wang, Shuzhe, et al. "Dust3r: Geometric 3d vision made easy." Proceedings of the IEEE/CVF Conference on Computer Vision and Pattern Recognition. 2024.
> [4] Yang, Lihe, et al. "Depth anything: Unleashing the power of large-scale unlabeled data." Proceedings of the IEEE/CVF Conference on Computer Vision and Pattern Recognition. 2024.
> [5] Ge, Yixiao, et al. "Self-supervising fine-grained region similarities for large-scale image localization." Computer Vision–ECCV 2020: 16th European Conference, Glasgow, UK, August 23–28, 2020, Proceedings, Part IV 16. Springer International Publishing, 2020.
> [6] Ali-bey, Amar, Brahim Chaib-draa, and Philippe Giguere. "Gsv-cities: Toward appropriate supervised visual place recognition." Neurocomputing 513 (2022): 194-203.
> ```
> > *Question*: If the above question is answerd and concerns addressed, I'm happy to raise my rating.
>
>  **We sincerely appreciate your constructive suggestions and believe that the additional experiments, analysis, and explanations significantly improve the quality of our submission.**
>
> **We hope that this provides sufficient reasons to raise the score.**

---

> ### Author Response · Authors · 2024-11-19
>
> Dear Review ZnHW,
>
> Thank you again for the great efforts and valuable comments. We have carefully addressed the main concerns in detail. We hope you might find the response satisfactory. As the discussion phase is about to close, we are very much looking forward to hearing from you about any further feedback. We will very happy to clarify any further concers (if any).
>
> Best, Authors

---

> ### Comment · Reviewer_ZnHW · 2024-11-21
>
> The aurhors have comprehensively addressed my concerns and questions.
>
> With the extra experiment and insights provided, I believe the claims in the paper are well supported.
>
> If the authors could add the insights of trilet loss and PCA to the final version of the paper, I believe the quality will be largely improved.
>
> I also suggest the authors add a legnd to Figure 1 so that the explanation in the discussion can be well reflected in the paper.
>
> I have updated my score.

---

> ### Author Response · Authors · 2024-11-22
>
> Thank you for your positive feedback and for updating us on the score.
>
> We genuinely appreciate your insightful and constructive comments.
>
> We are also greatful for the multiple rounds of communication, as your engagement has greatly enhanced the quality of our work.

---

### Author Response · Authors · 2024-11-25

**Dear Reviewers and Area Chair,**

We are delighted to see the strengths of our paper recognized:

- "the organization and presentation of this paper are good" (**YFv3**, snqV),

- "**simple and elegant**" (snqV) — this perfectly aligns with our research philosophy, and we deeply appreciate this comment,

- "valuable for the community" (ZnHW, snqV),

- “thoroughly evaluated (ablated)” (ZnHW, **YFv3**, snqV),

- “thoroughly presents the historical overview of the VPR field” (fhKJ).

|              | fhKJ | snqV | YFv3 | ZnHW |
|--------------|:------:|:------:|:------:|:------:|
| Soundness | 2    | 3   | 3  | 3   |
| Presentation | 1    | 3   | 3  | 2   |
| Contribution | 2    | 3   | 2  | 3   |
| Init. Rating | 3    | 6   | 3  | 3   |
| Rating | 5 | 8 |  ?  | 6   |

We have carefully addressed **all concerns** raised in our rebuttal and are open to further discussion.

Your thoughtful feedback has been instrumental in refining our work.

In response to your comments, we have incorporated additional analyses and experiments into the updated version of our paper.

We plan to further enhance the paper before the official announcement of results.

Thank you for your time and insightful input!

**Best regards,**

*The Authors*

---

### Note · Authors · 2025-01-23

**Comment:**

**Dear Reviewers, ACs, and SACs,**

Thank you for taking the time to review our manuscript and for providing valuable feedback. Your insights have been immensely helpful in identifying areas where we can further enhance the clarity, impact, and utility of our work.

I believe the following points require clarification:

```
In particular, dimensionality compression (in the context of NetVLAD features) was discussed in [A, B], while the Generalized Channel Attention (GCA) was described as a baseline work that performs feature fusion [C].
```

1. This comment is **entirely incorrect**, as we have already addressed this issue during the discussions. In our initial submission, we compared our method with those in [A, B], but we cited an earlier reference (2016). As for [C], GCA corresponds to $w \times x$, while Feature Fusion refers to $x_1 + x_2 + x_3$.

2. All discussions are conducted under options visible to ACs, SACs, and PCs, while the ICLR system provides reply options visible only to certain reviewers.

3. Given that our scores are 8, 6, and 3, our response to reviewer fhKJ’s comment is of a neutral nature. Our primary consideration is that irrelevant information in emails would obviously disturb other reviewers.

4. To be honest, from reviewer fhKJ’s initial review comments, I sensed harshness (S1), malice (W1, W3), carelessness (Q1, Q2, Q5, Q7), and sarcasm (Q8).

Thank you once again for your time and thoughtful input.

*Authors*

**Withdrawal Confirmation:**

I have read and agree with the venue's withdrawal policy on behalf of myself and my co-authors.